# Regularized Softmax Deep Multi-Agent $Q$-Learning

Ling Pan[1], Tabish Rashid[2], Bei Peng[3],*Longbo Huang[1], Shimon Whiteson[2]
[1]Institute for Interdisciplinary Information Sciences, Tsinghua University
pl17@mails.tsinghua.edu.cn, longbohuang@tsinghua.edu.cn
[2]University of Oxford
tabish.rashid@cs.ox.ac.uk, shimon.whiteson@cs.ox.ac.uk
[3]University of Liverpool
bei.peng@liverpool.ac.uk

## Abstract

Tackling overestimation in $Q$-learning is an important problem that has been extensively studied in single-agent reinforcement learning, but has received comparatively little attention in the multi-agent setting. In this work, we empirically demonstrate that QMIX, a popular $Q$-learning algorithm for cooperative multi-agent reinforcement learning (MARL), suffers from a more severe overestimation in practice than previously acknowledged, and is not mitigated by existing approaches. We rectify this with a novel regularization-based update scheme that penalizes large joint action-values that deviate from a baseline and demonstrate its effectiveness in stabilizing learning. Furthermore, we propose to employ a softmax operator, which we efficiently approximate in a novel way in the multi-agent setting, to further reduce the potential overestimation bias. Our approach, Regularized Softmax (RES) Deep Multi-Agent $Q$-Learning, is general and can be applied to any $Q$-learning based MARL algorithm. We demonstrate that, when applied to QMIX, RES avoids severe overestimation and significantly improves performance, yielding state-of-the-art results in a variety of cooperative multi-agent tasks, including the challenging StarCraft II micromanagement benchmarks.

## 1 Introduction

In recent years, multi-agent reinforcement learning (MARL) has achieved significant progress [4, 20] under the popular centralized training with decentralized execution (CTDE) paradigm [23, 15, 7]. In CTDE, the agents must learn decentralized policies so that at execution time they can act based on only local observations, but the training itself is centralized, with access to global information. A critical challenge in this setting is how to represent and learn the joint action-value function [28].

In learning the value function, overestimation is an important challenge that stems from the max operator [34] typically used in the bootstrapping target. Specifically, the max operator in $Q$-learning [39] approximates the maximum *expected* value with the maximum *estimated* value. This can lead to overestimation as $\mathbb{E}\left[\max_i X_i\right] \geq \max_i \mathbb{E}\left[X_i\right]$ due to noise [34, 11], where $X_i$ is a random variable representing the $Q$-value of action $i$ given a state. This overestimation error can accumulate during learning, lead to sub-optimal policy updates and behaviors, and hurt the performance of both value-based [37, 2, 32, 16] and actor-critic algorithms [8], and has been widely studied in the single-agent domain. However, overestimation can be even more severe in the multi-agent setting. For example, suppose there are $n$ agents, each agent has $K$ actions, and the $Q$-value for each action given a state is independently drawn from a uniform distribution $U(0, 1)$. Then, $\max_i \mathbb{E}\left[X_i\right]$ is $\frac{1}{2}$ while $\mathbb{E}\left[\max_i X_i\right]$ is $\frac{K^n}{K^n+1}$, which quickly approaches 1 (the maximum value of $X_i$) as the size of the joint action space

---

*Work done while at University of Oxford.

35th Conference on Neural Information Processing Systems (NeurIPS 2021).

increases exponentially with the number of agents. Nonetheless, this problem has received much less attention in MARL.

QMIX [29] is a popular CTDE deep multi-agent $Q$-learning algorithm for cooperative MARL. It combines the agent-wise utility functions $Q_a$ into the joint action-value function $Q_{tot}$, via a monotonic mixing network to ensure consistent value factorization. Due to its superior performance, there have been many recent efforts to improve QMIX's representation capability [31, 41, 27, 38]. However, the overestimation problem of the joint-action $Q$-function $Q_{tot}$ can be exacerbated in QMIX due not only to overestimation in agents' $Q_a$ but also the non-linear monotonic mixing network. Despite this, the role of overestimation in limiting its performance has been largely overlooked.

In this paper, we show empirically that overestimation in deep multi-agent $Q$-learning is more severe than previously acknowledged and can lead to divergent learning behavior in practice. In particular, consider double estimators, which can successfully reduce overestimation bias in the single-agent domain [37, 8]. Although QMIX applies Double DQN [37] to estimate the value function (as mentioned in Appendix D.3 in [28] and open-source PyMARL [30] implementations), we find that it is ineffective in deep multi-agent $Q$-learning. As shown in Figure 1 (same experimental setup as in Section 5), value estimates of the joint-action $Q$-function can increase without bound in tasks from the

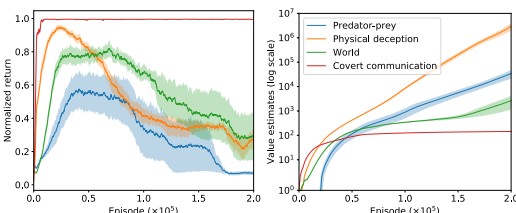

Figure 1: Normalized performance (left) and value estimations in log scale (right) of QMIX in the multi-agent particle environments. Value estimations can grow unbounded (right) and lead to catastrophic performance degradation (left).

multi-agent particle framework [18], yielding catastrophic performance degradation. Our experiments show that surprisingly, even applying Clipped Double $Q$-learning (a key technique from a state-of-the-art TD3 [8] algorithm) to the multi-agent setting does not resolve the severe overestimation bias in the joint-action $Q$-function. Therefore, alleviating overestimation in MARL is a particularly important and challenging problem.

To tackle this issue, we propose a novel update scheme that penalizes large joint-action $Q$-values. Our key idea is to introduce a regularizer in the Bellman loss. A direct penalty on the magnitude of joint-action $Q$-values can result in a large estimation bias and hurt performance. Instead, to better trade off learning efficiency and stability, we introduce a baseline into the penalty, thereby constraining the joint-action $Q$-values to not deviate too much from this baseline. Specifically, we use the discounted return as the baseline. By regularizing towards a baseline, we stabilize learning and effectively avoid the unbounded growth in our value estimates.

However, regularization is not enough to fully avoid overestimation bias in the joint-action $Q$-function due to the max operator in the target's value estimate [34]. To this end, we propose to employ a softmax operator, which has been shown to efficiently improve value estimates in the single-agent setting [32, 26, 25]. Unfortunately, a direct application of the softmax operator is often too computationally expensive in the multi-agent case, due to the exponentially-sized joint action space. We therefore propose a novel method that provides an efficient and reliable approximation to the softmax operator based on a joint action subspace, where the gap between our approximation and its direct computation converges to $0$ at an exponential rate with respect to its inverse temperature parameter. The computational complexity of our approximation scales only linearly in the number of agents, as opposed to exponentially for the original softmax operator. We show that our softmax operator can further improve the value estimates in our experiments. We refer to our method as *RES (Regularized Softmax) deep multi-agent $Q$-learning*, which utilizes the discounted return-based regularization and our approximate softmax operator.

To validate RES, we first prove that it can reduce the overestimation bias of QMIX. Next, we conduct extensive experiments in the multi-agent particle tasks [18], and show that RES simultaneously enables stable learning, avoids severe overestimation when applied to QMIX, and achieves state-of-the-art performance. RES is not tied to QMIX and can significantly improve the performance and stability of other deep multi-agent $Q$-learning algorithms, e.g., Weighted-QMIX [27] and QPLEX [38], demonstrating its versatility. Finally, to demonstrate its ability to scale to more complex scenarios, we evaluate it on a set of challenging StarCraft II micromanagement tasks [30]. Results show that RES-QMIX provides a consistent improvement over QMIX in all scenarios tested.

## 2 Background

**Decentralized partially observable Markov decision process (Dec-POMDP).** A fully cooperative multi-agent task can be formulated as a Dec-POMDP [22] represented by a tuple $\langle A, S, U, P, r, Z, O, \gamma \rangle$, where $A \equiv \{1, ..., n\}$ denotes the finite set of agents, $s \in S$ is the global state, and $\gamma \in [0, 1)$ is the discount factor. The action space for an agent is $U$ with size $K$. At each timestep, each agent $a \in A$ receives an observation $z \in Z$ from the observation function $O(s, a)$ due to partial observability, and chooses an action $u_a \in U$, which forms a joint action $\boldsymbol{u} \in \boldsymbol{U} \equiv U^n$. This leads to a transition to the next state $s' \sim P(s'|s, \boldsymbol{u})$ and a joint reward $r(s, \boldsymbol{u})$. We assume that the reward function is bounded [35], i.e., $|r(s, \boldsymbol{u})| \leq R_{\max}$. Each agent has an action-observation history $\tau_a \in T \equiv (Z \times U)^*$, based on which it constructs a policy $\pi_a(u_a|\tau_a)$. The goal is to find an optimal joint policy $\boldsymbol{\pi} = \langle \pi_1, ..., \pi_n \rangle$, whose joint action-value function is $Q^{\boldsymbol{\pi}}(s_t, \boldsymbol{u}_t) = \mathbb{E}[\sum_{i=0}^{\infty} \gamma^i r_{t+i}]$.

**Deep multi-agent $Q$-learning.** Deep multi-agent $Q$-learning [33, 29, 31, 38] extends deep $Q$-learning [39], a popular value-based method for learning optimal action values, to the multi-agent setting. Given transitions $(s, \boldsymbol{u}, r, s')$ sampled from the experience replay buffer $\mathcal{B}$, its objective is to minimize the mean squared error loss $\mathcal{L}(\theta)$ on the temporal-difference (TD) error $\delta = y - Q_{tot}(s, \boldsymbol{u})$, where $y = r + \gamma \max_{\boldsymbol{u}'} \bar{Q}_{tot}(s', \boldsymbol{u}')$ is the target value, and $\bar{Q}_{tot}$ is the target network for the joint-action $Q$-function that is periodically copied from $Q_{tot}$. Parameters of $Q_{tot}$ are denoted by $\theta$ that are updated by $\theta' = \theta - \alpha \nabla_\theta \mathcal{L}(\theta)$, where $\alpha$ is the learning rate. To mitigate the overestimation bias from the max operator in the target's value estimate [34], Double DQN [37, 11] estimates the target value as $y = r + \gamma \bar{Q}_{tot}(s', \arg\max_{\boldsymbol{u}'} Q_{tot}(s', \boldsymbol{u}'))$ which decouples action selection and evaluation.

**Centralized training with decentralized execution (CTDE).** In CTDE [23, 15], agents are trained in a centralized way with access to the overall action-observation history and global state during training, but during execution have access only to their own local action-observation histories. The individual-global-max (IGM) property [31] is a popular concept to realize efficient CTDE as in Eq. (1), where $Q_{tot}$ and $Q_a$ denote the joint-action $Q$-function and agent-wise utilities respectively.

$$\arg\max_{\boldsymbol{u}} Q_{tot}(s, \boldsymbol{u}) = \big( \arg\max_{u_1} Q_1(s, u_1), \cdots, \arg\max_{u_n} Q_n(s, u_n) \big). \tag{1}$$

The IGM property enables efficient decentralized execution, which ensures the consistency between greedy action selection in the local and joint $Q$-values. QMIX [29] is a popular CTDE method that combines $Q_a$ into $Q_{tot}$ via a non-linear monotonic function $f_s$ that is state-dependent as in Eq. (2). To satisfy the IGM property, $f_s$ is constrained to be monotonic in $Q_a$, i.e., $\frac{\partial f_s}{\partial Q_a} \geq 0, \forall a \in A$, which is achieved by enforcing non-negative weights in $f_s$.

$$Q_{tot}(s, \boldsymbol{u}) = f_s\big(Q_1(s, u_1), \cdots, Q_n(s, u_n)\big). \tag{2}$$

## 3 Overestimation in QMIX

In this section, we show empirically that the overestimation problem for deep multi-agent $Q$-learning can be more severe in practice than previously acknowledged. In particular, we demonstrate that state-of-the-art methods for tackling this issue in the single-agent domain can fail in our multi-agent setting.

We investigate the behavior of QMIX in the predator-prey task from the multi-agent particle environments [18], where 3 slower predators need to coordinate to capture a faster prey to solve the task. A detailed description of the task is included in Appendix E.1.1. We consider a fully cooperative setting where the prey is pre-trained by MADDPG [18] to avoid capture by the predators, and the predators need to learn to cooperate with each other in order to surround and capture the prey.

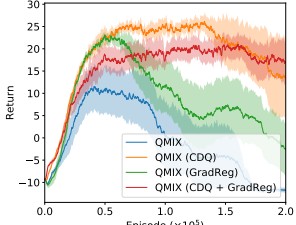

Figure 2: Return of QMIX and its variants in the predator-prey environment.

Figures 2 and 3(a) show the performance of QMIX and its estimated values of the joint-action $Q$-function during training respectively. QMIX suffers from catastrophic performance degradation and a severe overestimation bias, where its value estimates grow without bound. Figures 3(b) and 3(c) illustrate the mean agent-wise utilities $Q_a$ and their gradients $\partial f_s / \partial Q_a$ over agents respectively. We see that for QMIX, the gradients $\partial f_s / \partial Q_a$ increase rapidly and continuously during learning. Figure 3(d) shows the learned weight (matrix) of the monotonic mixing network $f_s$. The weights similarly

grow larger during training and further amplify the overestimation in each $Q_a$ when computing $Q_{tot}$, leading to a severely overestimated joint-action $Q$-function that continues to grow without bound.

To reduce overestimation, Ackermann et al. [1] extend the state-of-the-art TD3 algorithm [8], which addresses overestimation in the single-agent case, to the multi-agent setting. We obtain QMIX (CDQ) by applying the key Clipped Double $Q$-learning technique from TD3 [8] to per-agent utilities in QMIX for value estimation.[2] As shown in Figure 3(b), QMIX (CDQ) slows the increase of agent-wise utilities. However, Figure 3(a) shows that it still does not eliminate the severe overestimation bias in the joint-action $Q$-function, and subsequently suffers a performance degradation as shown in Figure 2. This is due to the gradients of the monotonic mixing network, $\partial f_s / \partial Q_a$, which still continuously increase as shown in Figure 3(c), leading to the large and increasing value estimates in Figure 3(a).

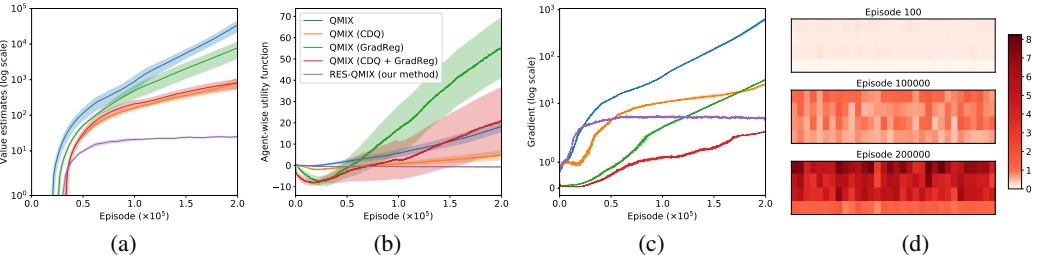

Figure 3: Learning statistics during training in predator-prey. (a) Value estimates in log scale. (b) Mean agent-wise utility functions $Q_a$ over agents. (c) Mean gradients $\frac{\partial f_s}{\partial Q_a}$ over agents in log scale. (d) The learned weight (matrix) in the monotonic mixing network $f_s$ of QMIX, where a darker color represents a larger value (a detailed description is in Appendix A.1).

Another way to avoid overly large value estimates is to limit the gradients themselves. We propose Gradient Regularization (GradReg) to prevent the gradients $\partial f_s / \partial Q_a$ from growing too large by regularizing the gradient using a quadratic penalty. Specifically, the loss function of QMIX (GradReg) is defined as $\mathcal{L}_{\text{GradReg}}(\theta) = \mathbb{E}_{(s, \boldsymbol{u}, r, s') \sim \mathcal{B}} \left[ \delta^2 + \lambda (\partial f_s / \partial Q_a)^2 \right]$, where $\delta$ is the TD error defined in Section 2 and $\lambda$ is the regularization coefficient. However, although QMIX (GradReg) with a tuned coefficient for the regularizer prevents the gradients from growing overly large during the early stage of learning (Figure 3(c)), the agent-wise utility functions grow larger instead as shown in Figure 3(b). This then still leads to a particularly large value estimate as shown in Figure 3(a). Finally, a direct combination of both CDQ and GradReg also fails to avoid the problem.

From these experiments, we see that even applying the state-of-the-art TD3 [8] method from the single-agent literature can fail in our multi-agent setting. In addition, it is also insufficient to tackle the problem by regularizing the magnitude of the gradients $\partial f_s / \partial Q_a$. These results show that mitigating overestimation in MARL is particularly important and challenging, and requires novel solutions.

## 4  Regularized Softmax (RES) Deep Multi-Agent $Q$-Learning

Our analysis shows that preventing gradients from being large as in QMIX (GradReg) and/or utilizing clipped double estimators as in QMIX (CDQ) are not sufficient to prevent overly large value estimates. It is important to note that these methods take an *indirect* approach to reduce the overestimation of the joint-action $Q$-value, and as shown do not solve the severe overestimation problem. Therefore, we propose a novel way to directly penalize large joint-action $Q$-values to avoid value estimate explosion.

Directly penalizing large joint-action $Q$-values can push their values towards 0, yielding a large estimation bias. Instead, we introduce a baseline into the penalty to better trade off learning efficiency and stability. Specifically, the new learning objective penalizes joint-action $Q$-values deviating from a baseline $b(s, \boldsymbol{u})$, by adding a regularizer to the loss: $\lambda \left( Q_{tot}(s, \boldsymbol{u}) - b(s, \boldsymbol{u}) \right)^2$, where we use the mean squared error loss and $\lambda$ is the regularization coefficient. A potential choice for the baseline is the $N$-step return, i.e., $b(s, \boldsymbol{u}) = \sum_{t=0}^{N-1} \gamma^t r_t + \gamma^N \max_{\boldsymbol{u}'} Q_{tot}(s_N, \boldsymbol{u}')$. Figure 4(a) shows the value estimates of Regularized (RE)-QMIX ($b = N$-step return) using the $N$-step return as the baseline (with best $N, \lambda$). However, as it still involves $Q_{tot}$, the value estimation can still grow extremely large despite being discounted, yielding estimates that almost coincide with those of QMIX.

---

[2]Applying CDQ on $Q_{tot}$ leads to larger value estimates than applying it on $Q_a$ (details are in Appendix A.2).

Therefore, we propose to use the discounted return starting from state $s$ as the baseline to regularize learning, i.e., $b(s, \boldsymbol{u}) = R_t(s, \boldsymbol{u}) = \sum_{k=0}^{\infty} \gamma^k r_{t+k}$ for the $t$-th timestep. The regularized learning objective directly penalizes large joint-action $Q$-values and allows the value estimates to remain grounded by real data. As shown in Figure 4(a), RE-QMIX effectively avoids particularly large value estimations and stabilizes learning.

However, while the regularized learning objective is effective in resolving value explosion, it can still overestimate due to the use of the max operator when computing target value estimates [34]. Figure 4(b) shows the estimated and true values of RE-QMIX (with best $\lambda$). The estimated values are computed by averaging over 100 states sampled from the replay buffer at each timestep, and we estimate true values by averaging the discounted returns which are obtained by following the greedy policy with respect to the current $Q_{tot}$ starting from the sampled states. As shown in Fig-

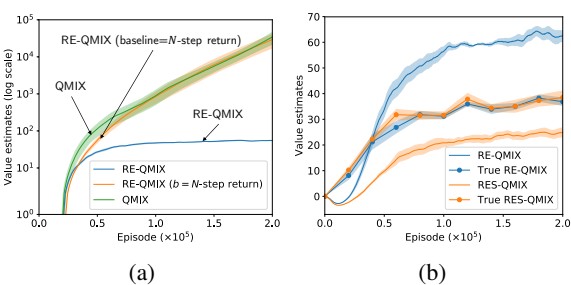

(a)               (b)

Figure 4: Comparison of value estimates in predator-prey. (a) Value estimation in log scale for QMIX and RE-QMIX with different baselines. (b) True values and estimated values of RE-QMIX and RES-QMIX.

ure 4(b), although RE-QMIX avoids unbounded growth of its value estimates, it does not fully avoid the overestimation bias, which still estimates the target value according to the max operator [34].

To further mitigate overestimation bias in the joint-action $Q$-function, we adopt the softmax operator, which has shown great potential in reducing overestimation bias in place of the max operator in single-agent domains [32, 26, 25]. Specifically, the softmax operator is defined in Eq. (3), where $\beta \geq 0$ is the inverse temperature parameter.

$$\text{sm}_{\beta, \boldsymbol{U}}\left(Q_{tot}(s, \cdot)\right) = \sum_{\boldsymbol{u} \in \boldsymbol{U}} \frac{e^{\beta Q_{tot}(s, \boldsymbol{u})}}{\sum_{\boldsymbol{u}' \in \boldsymbol{U}} e^{\beta Q_{tot}(s, \boldsymbol{u}')}} Q_{tot}(s, \boldsymbol{u}). \tag{3}$$

When $\beta$ approaches $\infty$ or 0, softmax reduces to the max or mean operator respectively. Unfortunately, computing Eq. (3) in the multi-agent case can be computationally intractable as the size of the joint action space grows exponentially with the number of agents. In addition, as the action space in the multi-agent case is much larger than that in the single-agent case, some joint-action $Q$-value estimates $Q_{tot}(s, \boldsymbol{u})$ can be unreliable due to a lack of sufficient training. Thus, directly taking them all into consideration for computing the softmax operator in Eq. (3) can result in inaccurate value estimates.

We propose a novel method to approximate the computation of the softmax operator efficiently and reliably. Specifically, we first obtain the maximal joint action $\hat{\boldsymbol{u}} = \arg\max_{\boldsymbol{u}} Q_{tot}(s, \boldsymbol{u})$ with respect to $Q_{tot}$ according to the individual-global-max (IGM) property discussed in Section 2. Then, for each agent $a$, we consider $K$ joint actions, by changing only agent $a$'s action while keeping the other agents' actions $\hat{\boldsymbol{u}}_{-a}$ fixed. We denote the resulting action set of agent $a$ by $U_a = \{(u_a, \hat{\boldsymbol{u}}_{-a}) | u_a \in U\}$. Finally, we form a joint action subspace $\hat{\boldsymbol{U}} = U_1 \cup \cdots \cup U_n$, where each $U_a$ contributes $K$ actions, and use $\hat{\boldsymbol{U}}$ in Eq. (3) for computing the approximate softmax operator. In Theorem 1, we show that the gap between our approximation in $\hat{\boldsymbol{U}}$ and its direct computation in the joint action space $\boldsymbol{U}$ converges to 0 at an exponential rate with respect to $\beta$. The proof can be found in Appendix B.1.

**Theorem 1.** *Let $\boldsymbol{u}^*$ and $\boldsymbol{u}'$ denote the optimal joint actions in $\boldsymbol{U}$ and $\boldsymbol{U} - \hat{\boldsymbol{U}}$ with respect to $Q_{tot}$, respectively. The difference between our approximate softmax operator and its direct computation in the whole action space satisfies: $\forall s \in \mathcal{S}, |\text{sm}_{\beta, \hat{\boldsymbol{U}}}(Q_{tot}(s, \cdot)) - \text{sm}_{\beta, \boldsymbol{U}}(Q_{tot}(s, \cdot))| \leq \frac{2R_{\max}}{1-\gamma} \frac{|\boldsymbol{U} - \hat{\boldsymbol{U}}|}{|\boldsymbol{U} - \hat{\boldsymbol{U}}| + \exp(\beta(Q_{tot}(s, \boldsymbol{u}^*) - Q_{tot}(s, \boldsymbol{u}')))}$, where $|\boldsymbol{U} - \hat{\boldsymbol{U}}|$ denotes the size of the set $\boldsymbol{U} - \hat{\boldsymbol{U}}$.*

**Discussion.** Since its computational complexity is linear instead of exponential in the number of agents, our approximation is feasible to compute even when the number of agents grows large, as opposed to the original softmax operator. According to the IGM property, $\hat{\boldsymbol{U}}$ consists of joint actions that are close to the maximal joint action $\hat{\boldsymbol{u}}$. Thus, it it more likely to contain joint actions with more accurate and reliable value estimates, especially when using $\epsilon$-greedy exploration, since they are more likely to be sampled and therefore trained. Due to this reduced reliance on *unreliable* joint-action $Q$-values, our approximate softmax operator actually outperforms its counterpart using

a direct computation in the joint action space $\boldsymbol{U}$, as well as a random sampling scheme, as shown in Section 5.1.3. The full algorithm for our approximate softmax is in Appendix B.2. As shown in Figure 4(b), RES (Regularized Softmax)-QMIX fully addresses the overestimation bias and achieves better value estimates than RE-QMIX.[3]

The loss of Regularized Softmax (RES) Deep Multi-Agent $Q$-Learning $\mathcal{L}_{\text{RES}}(\theta)$ is defined as:

$$\mathbb{E}_{(s,\boldsymbol{u},r,s')\sim\mathcal{B}}\left[\delta_{\text{sm}_\beta}^2 + \lambda\left(Q_{tot}(s,\boldsymbol{u}) - R_t(s,\boldsymbol{u})\right)^2\right], \tag{4}$$

where $\delta_{\text{sm}_\beta} = r + \gamma\text{sm}_{\beta,\hat{U}}(\bar{Q}_{tot}(s',\cdot)) - Q_{tot}(s,\boldsymbol{u})$, and $\lambda$ denotes the coefficient of the regularizer.

To better understand the effectiveness of our method, we provide a theoretical justification that connects the loss function of RES and a new Bellman operator in Theorem 2.

**Theorem 2.** *Given the same sample distribution, the update of the RES method is equivalent to the update using $\mathcal{L}(\theta) = \mathbb{E}_{(s,\boldsymbol{u},r,s')\sim\mathcal{B}}\left[(y - Q_{tot}(s,\boldsymbol{u}))^2\right]$ with learning rate $(\lambda+1)\alpha$, which estimates the target value according to $y = \frac{r+\gamma\text{sm}_{\beta,\hat{U}}(\bar{Q}_{tot}(s',\cdot))}{\lambda+1} + \frac{\lambda R_t(s,\boldsymbol{u})}{\lambda+1}$.*

Its proof is in Appendix C. In the special case where $\lambda = 0$, it is equivalent to changing the max operator to our approximate softmax in the target value estimation. When $\lambda > 0$, it can be thought of as learning with a weighted combination of the 1-step TD target with our approximate softmax and the discounted return, which allows the bootstrapping target to remain grounded by real data.[4]

Let $\mathcal{T}$ be the value estimation operator which estimates the value of the next state $s'$. From Theorem 2, we have $\mathcal{T}_{\text{QMIX}} = \max_{\boldsymbol{u}'}\bar{Q}_{tot}(s',\boldsymbol{u}')$, $\mathcal{T}_{\text{RE-QMIX}} = \frac{\max_{\boldsymbol{u}'}\bar{Q}_{tot}(s',\boldsymbol{u}')+\lambda R_{t+1}(s')}{\lambda+1}$, and $\mathcal{T}_{\text{RES-QMIX}} = \frac{\text{sm}_{\beta,\hat{U}}(\bar{Q}_{tot}(s',\cdot))+\lambda R_{t+1}(s')}{\lambda+1}$. We now analyze the relationship between the value estimation bias of these operators in Theorem 3 (the proof is in Appendix D), and show that RES-QMIX can reduce the overestimation bias when built upon QMIX. It is also worth noting that since RES is general, it can be readily applied to other deep multi-agent $Q$-learning algorithms, as investigated in Section 5.1.4.

**Theorem 3.** *Let $B(\mathcal{T}) = \mathbb{E}[\mathcal{T}(s')] - \max_{\boldsymbol{u}'}Q_{tot}^*(s',\boldsymbol{u}')$ be the bias of value estimates of $\mathcal{T}$ and the true optimal joint-action $Q$-function $Q_{tot}^*$. Given the same assumptions as in [37] for the joint-action $Q$-function, where there exists some $V_{tot}^*(s') = Q_{tot}^*(s',\boldsymbol{u}')$ for different joint actions, $\sum_{\boldsymbol{u}'}\left(\bar{Q}_{tot}(s',\boldsymbol{u}') - V_{tot}^*(s')\right) = 0$, and $\frac{1}{|\boldsymbol{U}|}\sum_{\boldsymbol{u}'}\left(\bar{Q}_{tot}(s',\boldsymbol{u}') - V_{tot}^*(s')\right)^2 = C$ $(C > 0)$ with $\bar{Q}_{tot}$ an arbitrary joint-action $Q$-function, then $B(\mathcal{T}_{\text{RES-QMIX}}) \leq B(\mathcal{T}_{\text{RE-QMIX}}) \leq B(\mathcal{T}_{\text{QMIX}})$.*

From the proof of Theorem 3 and Figure 4(a), we find that RE-QMIX can reduce the overestimation bias of QMIX following the same standard assumptions in [37, 32]. In addition, Theorem 3 also shows that the bias of value estimates of RES-QMIX is no larger than that of RE-QMIX, which validates our RES method.

# 5 Experiments

We conduct a series of experiments in the multi-agent particle tasks [18] to answer: (i) How much can our RES method improve over QMIX? (ii) How does RES-QMIX compare against state-of-the-art methods in performance and value estimates? (iii) How sensitive is RES to important hyperparameters and what is the effect of each component? (iv) Can RES be applied to other algorithms? We also evaluate our method on the challenging SMAC benchmark [30] to demonstrate its scalability. The code is publicly available at `https://github.com/ling-pan/RES`.

## 5.1 Multi-Agent Particle Environments

The predator-prey (PP) task was introduced in Section 3. Physical deception (PD) involves $2$ cooperating agents and $1$ adversary, whose goal is to reach a single target landmark from a total of two landmarks, while the adversary is unaware of the target. World (W) involves $4$ slower agents who must coordinate to catch $2$ faster adversaries that desire to eat food. In covert communication

---

[3]Note that applying the softmax operator on agent-wise utilities results in a larger underestimation bias and significantly underperforms RES, a detailed discussion is included in Appendix B.3.

[4]The result of QMIX with learning rate $(\lambda+1)\alpha$ is in Appendix C.1 (also suffers from performance drop).

(CC), one agent must send a private message to the other over a public channel with a private key, while an adversary tries to reconstruct the message without the key. In the above tasks, we consider a fully cooperative setting where the adversaries are pre-trained by MADDPG [18] for $10^4$ episodes.

We compare RES against state-of-the-art value factorization algorithms including VDN [33], QMIX [29], QTRAN [31], Weighted QMIX (including CW-QMIX and OW-QMIX) [27], and QPLEX [38] using the PyMARL [30] implementations and setup, and the popular actor-critic algorithm MADDPG [18]. For RES-QMIX, we fix the inverse temperature $\beta$ to be $0.05$ while the regularization coefficient $\lambda$ is selected based on a grid search over $\{1e-2, 5e-2, 1e-1, 5e-1\}$ as investigated in Section 5.1.3. Each algorithm is run with five random seeds, and is reported in mean $\pm$ standard deviation. A detailed description of the tasks and implementation details is in Appendix E.1.

### 5.1.1 Performance Comparison

We first investigate how much of a performance improvement RES-QMIX achieves over QMIX. Performance comparison in different environments is shown in Figure 5, and the mean normalized return averaged over different environments can be found in Appendix E.2. The results show that RES-QMIX significantly outperforms QMIX in performance and efficiency, and achieves stable learning behavior, without any catastrophic performance degradation. We then investigate the performance of RES-QMIX in comparison with state-of-the-art baselines. As shown in Figure 5, RES-QMIX significantly outperforms all baselines and achieves state-of-the-art performance in PP, PD, and W tasks. In the CC environment, RES-QMIX matches the best performers including VDN, QTRAN and QPLEX, while QMIX and Weighted QMIX suffer from oscillation. We also demonstrate the robust performance of RES-QMIX in stochastic environments in Appendix E.3.

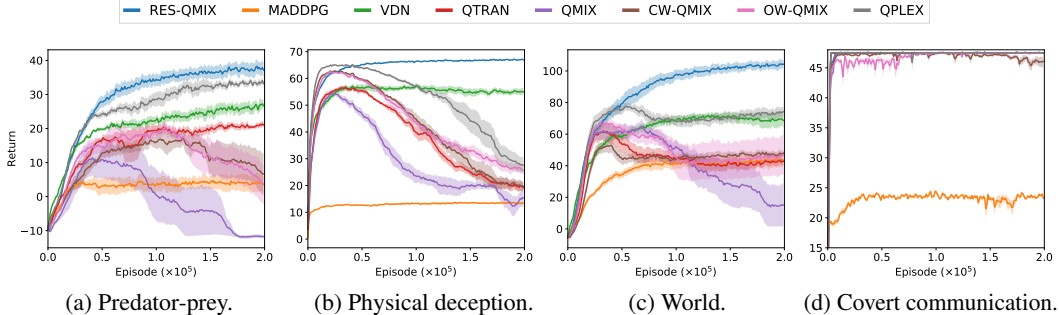

(a) Predator-prey.  (b) Physical deception.  (c) World.  (d) Covert communication.

Figure 5: Performance comparison.

### 5.1.2 Value Estimation Analysis

To understand the reasons for RES-QMIX's better performance, we analyze the bias of value estimation (the difference between estimated values and corresponding true values, which are computed in the same way as in Section 4) in predator-prey. To facilitate a fair comparison among different categories of algorithms including value-based and actor-critic methods, we normalize the bias of each method by $100 * \frac{\text{estimated value - true value}}{|\text{true value}|}$. Figure 6 shows that both QMIX and Weighted QMIX suffer from large and rapidly increasing overestimation bias, yielding the severe performance degradation shown in Figure 5(a). Value estimates of VDN, which is based on a linear decomposition of

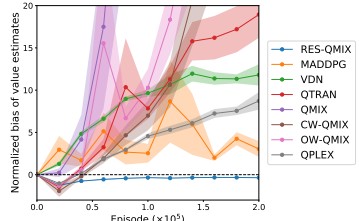

Figure 6: Normalized bias of value estimates.

$Q_{tot}$, increase more slowly at the end of training. However, it still incurs large overestimation as in QTRAN and QPLEX. Unlike all other value factorization methods, MADDPG learns an unfactored critic that directly conditions on the full state and joint action. It is less sample efficient, which indicates that value factorization is important in these tasks [14, 6, 13]. Thus, MADDPG results in a lower return and value estimates (shown in Appendix E.4) compared to all other value factorization methods, but still overestimates. RES-QMIX achieves the smallest bias and fully mitigates the overestimation bias of QMIX, resulting in stable performance and outperforming all other methods.

### 5.1.3 Ablation Study

We now analyze how sensitive RES-QMIX is to some important hyperparameters, and the effect of each component in our method including the regularizer, the softmax operator, and the approximation scheme in the predator-prey task. Full results for all environments can be found in Appendix E.5.

**The effect of the regularization coefficient** $\lambda$. Figure 7(a) shows the performance of RES-QMIX with varying $\lambda$. RES-QMIX is sensitive to this hyperparameter, which serves a critical role in trading off stability and efficiency. A small value of $\lambda$ fails to avoid the performance degradation, while a large value of $\lambda$ focuses more on learning based on the regularization term instead of the bootstrapping target, and affects learning efficiency. There exists an intermediate value that provides the best trade-off. In fact, we find that this is the only parameter that needs to be tuned for RES-QMIX.

**The effect of the inverse temperature** $\beta$. As shown in Figure 7(b), RES-QMIX is insensitive to the hyperparameter $\beta$, where the performance remains competitive for a wide range of $\beta$. This is different from previous studies [32, 26, 25] that are usually sensitive to the inverse temperature parameter, since they solely relies on the softmax operator for overestimation reduction (which is insufficient in the multi-agent setting as discussed in Section 3). We fix $\beta$ in RES-QMIX to be $0.05$ due to the relatively low sensitivity with respect to $\beta$, which performs the best, in all multi-agent particle environments.

**The effect of each component**. By comparing RES-QMIX against RE-QMIX and S-QMIX in Figure 7(c), we see that the regularization component is critical for stability, while combining with our softmax operator further improves learning efficiency.

**The effect of the approximation scheme**. We compare our proposed approximation scheme for the softmax operator with a random sampling scheme and a direct computation in the joint action space. Specifically, RES-QMIX (RS) randomly samples the same number of joint actions as in RES-QMIX, while RES-QMIX (DC) directly computes softmax in the joint action space. The runtime for QMIX, RES-QMIX, and RES-QMIX (DC) is $4.8$, $5.2$, and $10.7$ hours respectively, which shows that RES-QMIX only requires a small amount of extra computation compared to QMIX, and is much more computationally efficient than RES-QMIX (DC). Additionally, from Figure 7(c), it outperforms RES-QMIX (DC), showing that our approximation scheme is not only a necessary compromise for computational efficiency, but also improves performance as discussed in Section 4. RES-QMIX also significantly outperforms its counterpart with random sampling, which also validates its effectiveness.

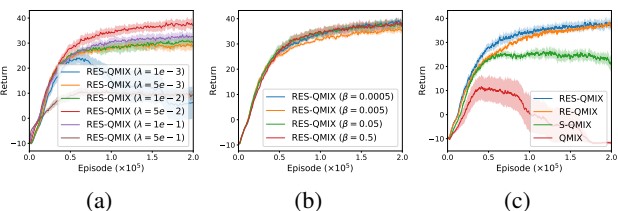

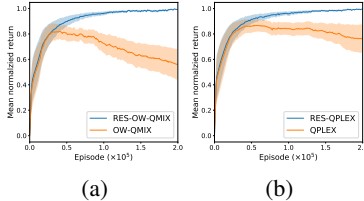

Figure 7: Ablation study. (a) Varying $\lambda$. (b) Varying $\beta$. (c) The effect of each component (the regularizer, the softmax operator, and the approximation scheme).

Figure 8: Mean normalized return of RES when applied to (a) Weighted QMIX and (b) QPLEX.

### 5.1.4 Applicability to Other Algorithms

The proposed RES method is general and can be readily applied to other $Q$-learning based MARL algorithms. To demonstrate its versatility, we apply it to two recent algorithms, Weighted QMIX [27] and QPLEX [38]. For Weighted QMIX, we use OW-QMIX as the base algorithm because it outperforms CW-QMIX as shown in Figure 5. The improvement in mean normalized return of RES-based methods over their vanilla counterparts in different environments is summarized in Figure 8, while the full comparison of learning curves in each environment is in Appendix E.6. As demonstrated, RES-Weighted QMIX and RES-QPLEX provide consistent improvement in performance and stability over Weighted QMIX and QPLEX, respectively, demonstrating the versatility of our RES method.

## 5.2 StarCraft II Micromanagement Benchmark

To demonstrate the ability of our method to scale to more complex scenarios, we also evaluate it on the challenging StarCraft II micromanagement tasks [30] by comparing RES-QMIX to QMIX using SC2 version 4.10 for five random seeds. Both algorithms are implemented using the PyMARL framework [30] with default hyperparameters as detailed in Appendix E.1. We evaluate the method on maps with different difficulties ranging from easy (`2s3z`, `3s5z`), hard (`2c_vs_64zg`), to super hard (`MMM2`) as classified in [30], with a detailed description in Appendix E.1.2.

Figures 9(a)-(d) show the test win rate. RES-QMIX provides a consistent performance improvement over QMIX. Additional comparison results with the most competitive baseline QPLEX, as in the multi-agent particle tasks, are provided in Appendix E.8, where RES-QMIX also outperforms QPLEX. Additionally, we show that RES is still effective even without double estimators. The results are presented in Appendix E.9, showing that RES-QMIX (single) outperforms both QMIX and QMIX (single) on all maps we tested.

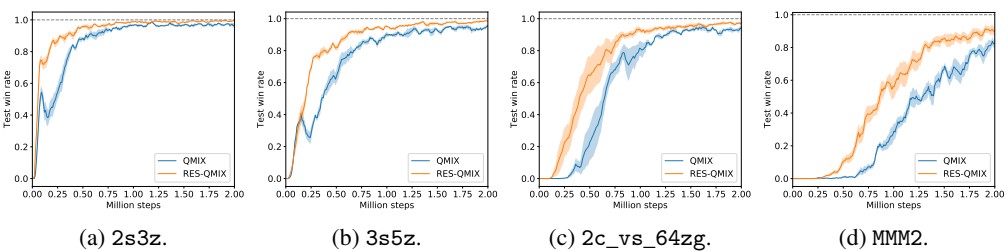

(a) 2s3z.     (b) 3s5z.     (c) 2c_vs_64zg.     (d) MMM2.

Figure 9: Comparison of RES-QMIX and QMIX in StarCraft II micromanagement tasks.

## 6 Related Work

Centralized training with decentralized execution (CTDE) [23, 15] is a popular paradigm in MARL, where a large number of recent work focuses on training agents under this paradigm. MADDPG [18] is an off-policy actor-critic algorithm that extends DDPG [17] to the multi-agent setting. Another line of research focuses on value-based $Q$-learning algorithm, which learns a centralized but factored joint action-value function $Q_{tot}$. For instance, VDN factorizes the joint-action $Q$-function into a linear combination of agent-wise utilities, and QMIX [29] improves the representation capability using a non-linear monotonic mixing network. There has also been a number of approaches improving QMIX's representation capability including QTRAN [31], Weighted QMIX [27], QPLEX [38] and its exploration ability [19, 10].

Methods to alleviate overestimation have been extensively studied in reinforcement learning, with a focus on the single-agent case [11, 37, 32, 16, 5, 8, 25]. However, as shown in [36], there is no unbiased estimator in general settings. Double $Q$-learning [11, 37] uses a pair of independent estimators to reduce overestimation in $Q$-learning. In [32, 26, 25], it is shown that the softmax operator is effective for reducing estimation bias in the single-agent setting. However, it is challenging to compute the softmax operator in the multi-agent case due to the exponentially-sized joint action space, for which we propose an efficient approximation scheme. Ackermann et al. [1] extend TD3 [8] to the multi-agent setting while Gan et al. [9] propose a soft Mellowmax operator [9] to tackle overestimation in MARL. However, the analysis in [9] is based on the assumption that the gradients $\partial f_s / \partial Q_a$ are bounded, which might not hold in practice as we have shown in Figure 3 (more technical discussion is in Appendix F). Both methods fail to tackle the severe overestimation problem as shown in Figure 2 and Appendix E.7.

To improve learning efficiency, He et al. [12] propose a learning objective based on lower and upper bounds of the optimal $Q$-function. As the upper bound is based on $N$-step returns, it cannot avoid the severe overestimation as investigated in Section 4. Self-imitation learning (SIL) [21] aims to improve exploration via leveraging the agent's past good experiences, which only performs SIL updates when the estimated value function is smaller than the discounted return. However, this is not effective for reducing overestimation bias in our case, as the SIL part pays little attention to experiences whose

estimated values are greater than the discounted return.[5] The update rule of RES shown in Theorem 2 is related to the mixed Monte Carlo update [3, 24]. However, our primary motivation is to reduce severe overestimation as opposed to speed up learning.

# 7 Conclusion

Overestimation is a critical problem in RL, and has been extensively studied in the single-agent setting. In this paper, we showed that it presents a more severe practical challenge in MARL than previously acknowledged, and solutions in the single-agent domain fail to successfully tackle this problem. We proposed the RES method based on a novel regularization-based update and the softmax operator with an efficient and reliable approximation in a novel way. Extensive experiments showed that RES-QMIX significantly reduces overestimation and outperforms state-of-the-art baselines. RES is general and can be applied to other deep multi-agent $Q$-learning methods, and can also scale to a set of challenging StarCraft II micromanagement tasks. The analysis and development of RES shed light on how to design better value estimation in MARL. Interesting directions for future work include theoretical study for the phenomenon discovered in Section 3, an adaptive scheduling for the regularization coefficient (e.g., meta-gradient approaches [40]), and the application of RES to other MARL methods.

## Acknowledgments and Disclosure of Funding

The work of Ling Pan and Longbo Huang is supported in part by the Technology and Innovation Major Project of the Ministry of Science and Technology of China under Grant 2020AAA0108400 and 2020AAA0108403. Ling Pan is supported by Microsoft Research Asia Fellowship. Tabish Rashid is supported by an EPSRC grant (EP/M508111/1, EP/N509711/1). This project has received funding from the European Research Council (ERC), under the European Union's Horizon 2020 research and innovation programme (grant agreement number 637713). We thank Qingpeng Cai and Tarun Gupta for insightful discussion.

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
