# Appendix for Regularized Softmax Deep Multi-Agent $Q$-Learning

**Ling Pan**[1], **Tabish Rashid**[2], **Bei Peng**[3],* **Longbo Huang**[1], **Shimon Whiteson**[2]
[1]Institute for Interdisciplinary Information Sciences, Tsinghua University
pl17@mails.tsinghua.edu.cn, longbohuang@tsinghua.edu.cn
[2]University of Oxford
tabish.rashid@cs.ox.ac.uk, shimon.whiteson@cs.ox.ac.uk
[3]University of Liverpool
bei.peng@liverpool.ac.uk

## A    Details of Results in Section 3 and Additional Results

### A.1    Description of Figure 3(d) in the Main Text

Figure 3(d) in the main text shows the learned weights in the monotonic mixing network $f_s$ of QMIX during learning, where the details of the network structure can be found in Section E.1. The weights of the monotonic mixing network are obtained from hypernetworks [5], whose outputs are reshaped into a matrix with appropriate size. Specifically, the outputs of the first and second layers are reshaped to (num_agents, embed_dim) and (embed_dim, 1) respectively, which corresponds to the first three rows (as there are three agents in the environment) and the final row in Figure 3(d) in the main text with embed_dim $= 32$ corresponding to the columns. A darker color represents a larger value.

### A.2    Additional Results of QMIX (CDQ)

In QMIX (CDQ), we apply CDQ on agent-wise utility functions $Q_a$ in QMIX by $\min\left(\bar{Q}_a(s', u'_a), Q_a(s', u'_a)\right)$, where $u'_a = \arg\max_{u'_a} Q_a(s', u'_a)$. Comparison results between applying CDQ on $Q_a$ and the joint-action $Q$-function $Q_{tot}$ (QMIX (CDQ-joint)) are shown in Figures 1(a)-(b). As demonstrated, QMIX (CDQ-joint) underperforms QMIX (CDQ), which also results in larger value estimates.

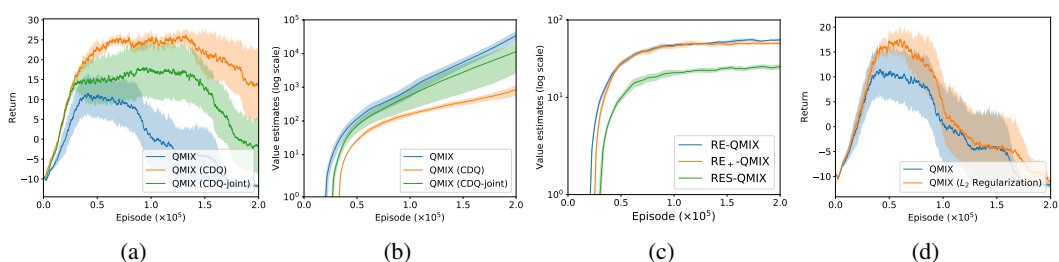

Figure 1: (a) and (b) Comparison of performance and value estimates of QMIX (CDQ) and QMIX (CDQ-joint). (c) Value estimates (in log scale) of RE-QMIX, RE$_+$-QMIX, and RES-QMIX.(d) Performance comparison of QMIX and QMIX ($L_2$ Regularization).

---

*Work done while at University of Oxford.

35th Conference on Neural Information Processing Systems (NeurIPS 2021).

### A.3 Additional Results of a Clipped Version of RE-QMIX

Figure 1(c) shows comparison results for value estimates of RE-QMIX, $RE_+$-QMIX, and RES-QMIX. Specifically, $RE_+$ uses a clipped version of the regularizer in RE-QMIX by $\lambda((Q_{tot}(s, \boldsymbol{u}) - R_t(s, \boldsymbol{u}))_+)^2$, where $\lambda$ denotes the coefficient and $(\cdot)_+ = \max(\cdot, 0)$. As a result, $RE_+$-QMIX only penalizes joint-action $Q$-values when $Q_{tot}(s, \boldsymbol{u}) > R_t(s, \boldsymbol{u})$. As shown in Figure 1(c), the value estimates for $RE_+$-QMIX are very close to those of RE-QMIX. This is because the regularizer is active in most cases ($94.4\%$ on average during training).

### A.4 Additional Results of QMIX ($L_2$ Regularization)

Figure 1(d) shows the performance comparison of QMIX and QMIX ($L_2$ Regularization) with best regularization coefficient. The loss function of QMIX ($L_2$ Regularization) is obtained by adding $\lambda||\theta||_2^2$ to the original loss function [8], where $\lambda$ is the coefficient. As shown, applying $L_2$ regularization fails to avoid performance degradation.

## B Our Approximate Softmax Operator

### B.1 Proof of Theorem 1

**Theorem 1.** *Let $\boldsymbol{u}^*$ and $\boldsymbol{u}'$ denote the optimal joint actions in $\boldsymbol{U}$ and $\boldsymbol{U} - \hat{\boldsymbol{U}}$ with respect to $Q_{tot}$, respectively. The difference between our approximate softmax operator and its direct computation in the whole action space satisfies: $\forall s \in \mathcal{S}, |\text{sm}_{\beta, \hat{\boldsymbol{U}}}(Q_{tot}(s, \cdot)) - \text{sm}_{\beta, \boldsymbol{U}}(Q_{tot}(s, \cdot))| \leq \frac{2R_{\max}}{1-\gamma} \frac{|\boldsymbol{U} - \hat{\boldsymbol{U}}|}{|\boldsymbol{U} - \hat{\boldsymbol{U}}| + \exp(\beta(Q_{tot}(s, \boldsymbol{u}^*) - Q_{tot}(s, \boldsymbol{u}')))}$, where $|\boldsymbol{U} - \hat{\boldsymbol{U}}|$ denotes the size of the set $\boldsymbol{U} - \hat{\boldsymbol{U}}$.*

*Proof.* By definition, we have that

$$|\text{sm}_{\beta, \hat{\boldsymbol{U}}}(Q_{tot}(s, \cdot)) - \text{sm}_{\beta, \boldsymbol{U}}(Q_{tot}(s, \cdot))| \tag{1}$$

$$= \left| \frac{\sum_{\boldsymbol{u} \in \boldsymbol{U}} \exp(\beta Q_{tot}(s, \boldsymbol{u})) Q_{tot}(s, \boldsymbol{u})}{\sum_{\bar{\boldsymbol{u}} \in \boldsymbol{U}} \exp(\beta Q_{tot}(s, \bar{\boldsymbol{u}}))} - \frac{\sum_{\boldsymbol{u} \in \hat{\boldsymbol{U}}} \exp(\beta Q_{tot}(s, \boldsymbol{u})) Q_{tot}(s, \boldsymbol{u})}{\sum_{\bar{\boldsymbol{u}} \in \hat{\boldsymbol{U}}} \exp(\beta Q_{tot}(s, \bar{\boldsymbol{u}}))} \right| \tag{2}$$

For simplicity, we denote $a = \sum_{\bar{\boldsymbol{u}} \in \hat{\boldsymbol{U}}} \exp(\beta Q_{tot}(s, \bar{\boldsymbol{u}}))$, $b = \sum_{\bar{\boldsymbol{u}} \in \boldsymbol{U} - \hat{\boldsymbol{U}}} \exp(\beta Q_{tot}(s, \bar{\boldsymbol{u}}))$, $c = \sum_{\boldsymbol{u} \in \hat{\boldsymbol{U}}} \exp(\beta Q_{tot}(s, \boldsymbol{u})) Q_{tot}(s, \boldsymbol{u})$, and $d = \sum_{\boldsymbol{u} \in \boldsymbol{U} - \hat{\boldsymbol{U}}} \exp(\beta Q_{tot}(s, \boldsymbol{u})) Q_{tot}(s, \boldsymbol{u})$. Then, Eq. (2) $= \frac{b}{a+b} \left| \frac{c}{a} - \frac{d}{b} \right|$.

We have that

$$\left| \frac{c}{a} - \frac{d}{b} \right| = |\text{sm}_{\beta, \hat{\boldsymbol{U}}}(Q_{tot}(s, \cdot)) - \text{sm}_{\beta, \boldsymbol{U} - \hat{\boldsymbol{U}}}(Q_{tot}(s, \cdot))| \tag{3}$$

$$\leq |\max_{\boldsymbol{u} \in \boldsymbol{U}} Q_{tot}(s, \boldsymbol{u}) - \min_{\boldsymbol{u} \in \boldsymbol{U}} Q_{tot}(s, \boldsymbol{u})| \tag{4}$$

$$\leq |\max_{\boldsymbol{u} \in \boldsymbol{U}} Q_{tot}(s, \boldsymbol{u})| + |\min_{\boldsymbol{u} \in \boldsymbol{U}} Q_{tot}(s, \boldsymbol{u})| \tag{5}$$

$$\leq 2 \max_{\boldsymbol{u} \in \boldsymbol{U}} |Q_{tot}(s, \boldsymbol{u})| \tag{6}$$

$$\leq \frac{2R_{\max}}{1 - \gamma}. \tag{7}$$

We also have that

$$\frac{b}{a+b} = \frac{\sum_{\bar{\boldsymbol{u}}\in\boldsymbol{U}-\hat{\boldsymbol{U}}}\exp(\beta Q_{tot}(s,\bar{\boldsymbol{u}}))}{\sum_{\bar{\boldsymbol{u}}\in\boldsymbol{U}}\exp(\beta Q_{tot}(s,\bar{\boldsymbol{u}}))} \tag{8}$$

$$\leq \frac{|\boldsymbol{U}-\hat{\boldsymbol{U}}|\exp(\beta Q_{tot}(s,\boldsymbol{u}'))}{|\boldsymbol{U}-\hat{\boldsymbol{U}}|\exp(\beta Q_{tot}(s,\boldsymbol{u}')) + \sum_{\bar{\boldsymbol{u}}\in\hat{\boldsymbol{U}}}\exp(\beta Q_{tot}(s,\bar{\boldsymbol{u}}))} \tag{9}$$

$$\leq \frac{|\boldsymbol{U}-\hat{\boldsymbol{U}}|\exp(\beta Q_{tot}(s,\boldsymbol{u}'))}{|\boldsymbol{U}-\hat{\boldsymbol{U}}|\exp(\beta Q_{tot}(s,\boldsymbol{u}')) + \exp(\beta Q_{tot}(s,\boldsymbol{u}^*))} \tag{10}$$

$$= \frac{|\boldsymbol{U}-\hat{\boldsymbol{U}}|}{|\boldsymbol{U}-\hat{\boldsymbol{U}}| + \exp(\beta(Q_{tot}(s,\boldsymbol{u}^*) - Q_{tot}(s,\boldsymbol{u}')))}. \tag{11}$$

Therefore, we obtain that

$$|\text{sm}_{\beta,\hat{\boldsymbol{U}}}(Q_{tot}(s,\cdot)) - \text{sm}_{\beta,\boldsymbol{U}}(Q_{tot}(s,\cdot))| \leq \frac{2R_{\max}}{1-\gamma}\frac{|\boldsymbol{U}-\hat{\boldsymbol{U}}|}{|\boldsymbol{U}-\hat{\boldsymbol{U}}| + \exp(\beta(Q_{tot}(s,\boldsymbol{u}^*) - Q_{tot}(s,\boldsymbol{u}')))}. \tag{12}$$

As a result, the gap converges to $0$ in an exponential rate with respect to the inverse temperature parameter $\beta$. □

## B.2 Algorithm and More Details for Computing the Approximate Softmax Operator

The full algorithm for computing the approximate softmax operator is in Algorithm 1.

---

**Algorithm 1** Approximate softmax operator

---

1: Obtain the maximal joint action w.r.t. $Q_{tot}$: $\hat{\boldsymbol{u}} = \arg\max_{\boldsymbol{u}} Q_{tot}(s,\boldsymbol{u})$
2: Construct $U_a$: $U_a = \{(u_a, \hat{\boldsymbol{u}}_{-a})|u_a \in U\}$
3: Construct the joint action subspace: $\hat{\boldsymbol{U}} = U_1 \cup \cdots \cup U_n$
4: Compute the approximate softmax operator:

$$\text{sm}_{\beta,\hat{\boldsymbol{U}}}(Q_{tot}(s,\cdot)) = \sum_{\boldsymbol{u}\in\hat{\boldsymbol{U}}} \frac{e^{\beta Q_{tot}(s,\boldsymbol{u})}}{\sum_{\boldsymbol{u}'\in\hat{\boldsymbol{U}}} e^{\beta Q_{tot}(s,\boldsymbol{u}')}} Q_{tot}(s,\boldsymbol{u})$$

---

Figure 2 also illustrates the computation of our approximate softmax operator in the joint action subspace $\hat{\boldsymbol{U}}$. The left part in Figure 2 corresponds to the maximal joint action w.r.t. $Q_{tot}$. The middle part demonstrates the joint action subspace $\hat{\boldsymbol{U}}$ (whose size is $nK$) for computing our approximate softmax operator, where the yellow block means that the corresponding action can be one of the actions in $U$. The right part shows the joint action space $\boldsymbol{U}$ (whose size is $K^n$), with the orange block showing that the action can be one of the actions in $U$ except for the local greedy action. As discussed in the main text, the action space in the multi-agent setting is much larger than that in the single-agent case, and some joint-action $Q$-value estimates $Q_{tot}(s,\boldsymbol{u})$ can be unreliable due to a lack of sufficient training. As a result, directly taking them all into consideration for computing the softmax operator as in the single-agent case [20, 14] can result in inaccurate value estimates. As shown in Figure 2, according to the individual-global-max (IGM) property discussed in Section 2 in the main text, $\hat{\boldsymbol{U}}$ consists of joint actions that are close to the maximal joint action $\hat{\boldsymbol{u}}$, which is more likely to contain joint actions with more accurate and reliable value estimates. As a result, our softmax operator provides an efficient and reliable approximation. Theorem 1 in the main text provides a theoretical guarantee for $\hat{\boldsymbol{U}}$, and Section 5.1.4 in the main text validates its effectiveness by comparing it with other choices.

## B.3 Discussion of the Softmax Operator on Agent-Wise Utilities

As discussed in the main text, we propose to employ the softmax operator to further mitigate the overestimation bias in the joint-action $Q$-function by $\text{sm}_{\beta,\hat{\boldsymbol{U}}}(Q_{tot}(s,\cdot))$. One may be interested

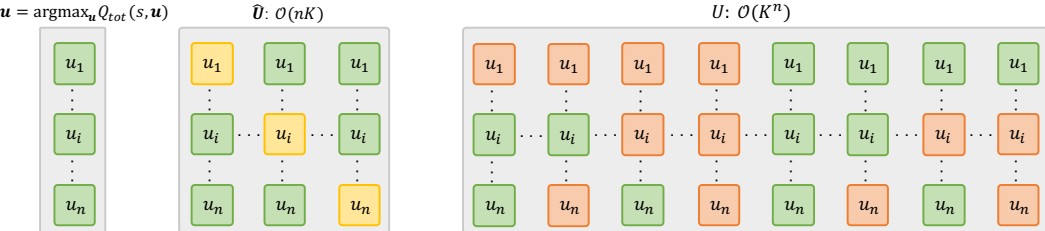

Figure 2: Left: The maximal joint action w.r.t. $Q_{tot}$. Middle: Illustration of the joint action subspace $\hat{U}$ for computing our approximate softmax operator. Right: Illustration of the joint action space $U$.

in its application on the agent-wise utility functions by $f_s\big(\mathrm{sm}_{\beta,U}(Q_1(s,\cdot)),\cdots,\mathrm{sm}_{\beta,U}(Q_n(s,\cdot))\big)$. The results are shown in Figure 3, where we refer to the method as RE-QMIX (softmax on $Q_a$). From Figure 3(b), we can see that this results in overly pessimistic value estimates and a larger underestimation bias shown in Figure 3(c). In addition, as shown in Figure 3(a), it also significantly underperforms RES-QMIX, demonstrating the necessity of a careful design of the softmax operator in deep multi-agent $Q$-learning methods in MARL.

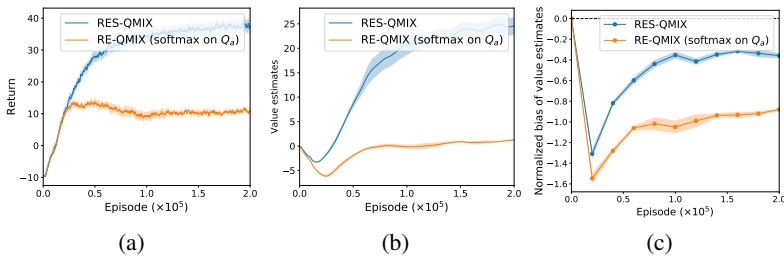

(a)  (b)  (c)

Figure 3: Comparison between RES-QMIX and RE-QMIX with the softmax operator on agent-wise utilities. (a) Performance. (b) Value estimates. (c) Normalized bias of value estimates.

## C   Proof of Theorem 2

**Theorem 2.** *Given the same sample distribution, the update of the RES method is equivalent to the update using $\mathcal{L}(\theta) = \mathbb{E}_{(s,\boldsymbol{u},r,s')\sim\mathcal{B}}\left[(y - Q_{tot}(s,\boldsymbol{u}))^2\right]$ with learning rate $(\lambda+1)\alpha$, which estimates the target value according to $y = \frac{r+\gamma\mathrm{sm}_{\beta,\hat{U}}(\bar{Q}_{tot}(s',\cdot))}{\lambda+1} + \frac{\lambda R_t(s,\boldsymbol{u})}{\lambda+1}$.*

*Proof.* Let $\theta$ and $\bar{\theta}$ denote parameters of $Q_{tot}$ and the target network $\bar{Q}_{tot}$ respectively. The gradient for the learning objective of the RES method is

$$\nabla_\theta\mathcal{L}_{\mathrm{RES}}(\theta) = -2(r + \gamma\mathrm{sm}_{\beta,\hat{U}}(\bar{Q}_{tot}(s',\cdot|\bar{\theta})) - Q_{tot}(s,\boldsymbol{u}|\theta))\nabla_\theta Q_{tot}(s,\boldsymbol{u}|\theta)$$
$$+ 2\lambda(Q_{tot}(s,\boldsymbol{u}|\theta) - R_t(s,\boldsymbol{u}))\nabla_\theta Q_{tot}(s,\boldsymbol{u}|\theta). \tag{13}$$

Then, we have

$$\theta' = \theta + 2\alpha(\lambda+1)\left(\frac{r+\gamma\mathrm{sm}_{\beta,\hat{U}}(\bar{Q}_{tot}(s',\cdot|\bar{\theta}))}{\lambda+1} + \frac{\lambda R_t(s,\boldsymbol{u})}{\lambda+1} - Q_{tot}(s,\boldsymbol{u}|\theta)\right)\nabla_\theta Q_{tot}(s,\boldsymbol{u}|\theta), \tag{14}$$

where $\alpha$ is the learning rate. Therefore, it is equivalent to using a learning rate $\alpha' = (\lambda+1)\alpha$, and estimating the target value $y$ by $\frac{r+\gamma\mathrm{sm}_{\beta,\hat{U}}(\bar{Q}_{tot}(s',\cdot))}{\lambda+1} + \frac{\lambda R_t(s,\boldsymbol{u})}{\lambda+1}$. $\square$

### C.1   Additional Results

Figure 4 shows the comparison results for QMIX, RES-QMIX, and QMIX with learning rate (lr) $(\lambda+1)\alpha$. We see that solely using a larger learning rate still fails to tackle the problem and cannot avoid performance degradation.

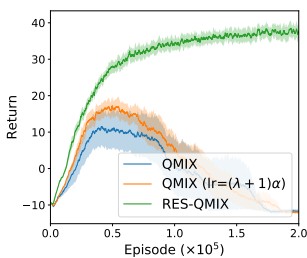

Figure 4: Comparison results of QMIX, QMIX (lr=$(\lambda + 1)\alpha$), and RES-QMIX.

# D Proof of Theorem 3

**Theorem 3** *Let $B(\mathcal{T}) = \mathbb{E}[\mathcal{T}(s')] - \max_{\boldsymbol{u}'} Q_{tot}^*(s', \boldsymbol{u}')$ be the bias of value estimates of $\mathcal{T}$ and the true optimal joint-action Q-function $Q_{tot}^*$. Given the same assumptions as in [21] for the joint-action Q-function, where there exists some $V_{tot}^*(s')$ such that $V_{tot}^*(s') = Q_{tot}^*(s', \boldsymbol{u}')$ for different joint actions, $\sum_{\boldsymbol{u}'} \left( \bar{Q}_{tot}(s', \boldsymbol{u}') - V_{tot}^*(s') \right) = 0$, and $\frac{1}{|\boldsymbol{U}|} \sum_{\boldsymbol{u}'} \left( \bar{Q}_{tot}(s', \boldsymbol{u}') - V_{tot}^*(s') \right)^2 = C$ ($C > 0$) with $\bar{Q}_{tot}$ an arbitrary joint-action Q-function, then $B(\mathcal{T}_{\text{RES-QMIX}}) \leq B(\mathcal{T}_{\text{RE-QMIX}}) \leq B(\mathcal{T}_{\text{QMIX}})$.*

*Proof.* For the left-hand-side, by definition, we have that

$$\text{sm}_{\beta, \hat{\boldsymbol{U}}} \left( \bar{Q}_{tot}(s', \cdot) \right) \leq \max_{\boldsymbol{u}' \in \hat{\boldsymbol{U}}} \bar{Q}_{tot}(s', \boldsymbol{u}') = \max_{\boldsymbol{u}' \in \boldsymbol{U}} \bar{Q}_{tot}(s', \boldsymbol{u}'). \tag{15}$$

Therefore, $\mathcal{T}_{\text{RES-QMIX}} \leq \mathcal{T}_{\text{RE-QMIX}}$, and $B(\mathcal{T}_{\text{RES-QMIX}}) \leq B(\mathcal{T}_{\text{RE-QMIX}})$.

For the right-hand-side, we have that

$$B(\mathcal{T}_{\text{RE-QMIX}}) - B(\mathcal{T}_{\text{QMIX}}) = \mathbb{E} \left[ \frac{\lambda}{\lambda + 1} \left( R_{t+1}(s') - \max_{\boldsymbol{u}' \in \boldsymbol{U}} \bar{Q}_{tot}(s', \boldsymbol{u}') \right) \right] \tag{16}$$

$$= \frac{\lambda}{\lambda + 1} \left( V_{tot}^\pi(s') - \mathbb{E} \left[ \max_{\boldsymbol{u}' \in \boldsymbol{U}} \bar{Q}_{tot}(s', \boldsymbol{u}') \right] \right) \tag{17}$$

$$\leq \frac{\lambda}{\lambda + 1} \left( \max_{\boldsymbol{u}' \in \boldsymbol{U}} Q_{tot}^*(s', \boldsymbol{u}') - \mathbb{E} \left[ \max_{\boldsymbol{u}' \in \boldsymbol{U}} \bar{Q}_{tot}(s', \boldsymbol{u}') \right] \right), \tag{18}$$

where $V_{tot}^\pi(s')$ is the expected discounted return starting from state $s'$ under the current behavior policy $\pi$, and Eq. (18) follows from the fact that its value is no larger than that of the optimal policy.

Then, it suffices to prove that $\mathbb{E} \left[ \max_{\boldsymbol{u}' \in \boldsymbol{U}} \bar{Q}_{tot}(s', \boldsymbol{u}') \right] \geq \max_{\boldsymbol{u}' \in \boldsymbol{U}} Q_{tot}^*(s', \boldsymbol{u}')$.

Assume that the joint-action $Q$-function follows the same assumptions of Theorem 1 in [21], i.e., there exists some $V_{tot}^*(s')$ such that $V_{tot}^*(s') = Q_{tot}^*(s', \boldsymbol{u}')$ for different joint actions, $\sum_{\boldsymbol{u}'} \left( \bar{Q}_{tot}(s', \boldsymbol{u}') - V_{tot}^*(s') \right) = 0$, and $\frac{1}{|\boldsymbol{U}|} \sum_{\boldsymbol{u}'} \left( \bar{Q}_{tot}(s', \boldsymbol{u}') - V_{tot}^*(s') \right)^2 = C$ for some $C > 0$. The assumption means that value estimates are correct on average, but there exists estimation error in the joint-action $Q$-value estimates for some joint actions. Then, following the analysis in [21], we have that $\max_{\boldsymbol{u}' \in \boldsymbol{U}} \bar{Q}_{tot}(s', \boldsymbol{u}') - \max_{\boldsymbol{u}' \in \boldsymbol{U}} Q_{tot}^*(s', \boldsymbol{u}') \geq 0$.

Therefore, RE-QMIX can reduce the overestimation bias of QMIX, and we have that $B(\mathcal{T}_{\text{RE-QMIX}}) \leq B(\mathcal{T}_{\text{QMIX}})$. $\qquad \square$

# E Experimental Details

## E.1 Experimental Setup

**Tasks.** The multi-agent particle environments [9] are based on an open-source implementation,[2] where the global state is the concatenation of observations of all agents. For StarCraft Multi-Agent

---

[2]https://github.com/shariqiqbal2810/multiagent-particle-envs

Challenge (SMAC),[3] we use the latest version 4.10. Note that the results reported in [18] use SC2.4.6.2.69232, and performance is not always comparable across versions.

**Baselines.** Value factorization methods including VDN, QMIX, and QTRAN are implemented using the PyMARL [18] framework,[4] while we use authors' open-source implementations for Weighted QMIX[5] and QPLEX.[6] The actor-critic method MADDPG is based on an open-source implementation,[7] which uses the Gumbel-Softmax trick to tackle discrete action spaces [9]. We use default hyperparameters and setup as in [18], where we list as in Table 1. The only exception is that we tune the target-update-interval for value factorization methods in multi-agent particle environments, as these algorithms fail to learn in these environments with default hyperparameters that are originally fine-tuned for SMAC environments. Specifically, the target-update-interval is 800 for QMIX, QTRAN, Weighted QMIX and QPLEX (which updates the target network every 800 episodes), while it remains 200 for VDN as the default value works well. Each agent network is a deep recurrent Q-network (DRQN) consisting of 3 layers: a fully-connected layer, a GRU layer with a $64$-dimensional hidden state, and a fully connected layer with ReLU activation. The mixing network consists of a 32-dimensional hidden layer using ELU activation, and the hypernetwork [5] consists of two layers with $64$-dimensional hidden state using ReLU activation. Note that QPLEX uses a different architecture for the mixing network with more parameters, which consists of two modules (with default hyperparameters as in [23]): a transformation network and a dueling mixing network with a multi-head attention module [22]. By default, all $Q$-learning based MARL algorithms (including VDN, QMIX, QTRAN, Weighted QMIX, QPLEX and our RES method) use double estimators as in Double DQN [21] to estimate the target value as pointed out in Appendix D.3 in [16]. All experiments are run on P100 GPU.

Table 1: Hyperparameters.

| Hyperparameter | Value |
|---|---|
| Discount factor | 0.99 |
| Replay buffer size | 5000 episodes |
| Batch size | 32 episodes |
| Warmup steps | 50000 |
| Optimizer | RMSprop |
| Learning rate | $5 \times 10^{-4}$ |
| Initial $\epsilon$ | 1.0 |
| Final $\epsilon$ | 0.05 |
| Linearly annealing steps for $\epsilon$ | 50k |
| Double DQN update | True |

**RES.** RES is also implemented based on the PyMARL [18] framework with the same network structures and hyperparameters as discussed above. For our RES method, to estimate the target joint-action $Q$-value using the softmax operator based on double estimators, the softmax weighting is computed in the joint action subspace based on current joint-action $Q$-network $Q_{tot}$ in Eq. (19):

$$\text{sm}_{\beta,\hat{U}}\left(\bar{Q}_{tot}(s,\cdot)\right) = \sum_{\boldsymbol{u}\in\hat{U}} \frac{e^{\beta Q_{tot}(s,\boldsymbol{u})}}{\sum_{\boldsymbol{u}'\in\hat{U}} e^{\beta Q_{tot}(s,\boldsymbol{u}')}} \bar{Q}_{tot}(s,\boldsymbol{u}). \tag{19}$$

As discussed in Section 5.1.3 in the main text, we only need to tune $\lambda$ while keeping $\beta$ fixed (whose performance is competitive within a wide range of values).

In multi-agent particle environments, for RES-QMIX, the inverse temperature $\beta$ of the softmax operator is fixed to be $0.05$, where the regularization coefficient $\lambda$ is selected based on a grid search over $\{1e-2, 5e-2, 1e-1, 5e-1\}$. Specifically, $\lambda$ is $1e-2$ for covert communication (CC), $5e-2$ for predator-prey (PP) and world (W), and $5e-1$ for physical deception (PD). As for RES-Weighted-QMIX, it is based on OW-QMIX [15] with an optimistic weighting, as it outperforms its counterpart

---

[3] https://github.com/oxwhirl/smac
[4] https://github.com/oxwhirl/pymarl
[5] https://github.com/oxwhirl/wqmix
[6] https://github.com/wjh720/QPLEX
[7] https://github.com/shariqiqbal2810/maddpg-pytorch

CW-QMIX as shown in Figure 5 in the main text. The parameter $\beta$ is $0.1$ while $\lambda$ is $5e - 2$ for PP and W, and $5e - 1$ for PD and CC. As for RES-QPLEX, we set $\beta = 0.1$ and $\lambda = 5e - 2$ for all environments. For RES-QMIX in SMAC, the regularization coefficient $\lambda$ is $1e - 2$ for 3m and the super hard map MMM2 while being $5e - 2$ for the remaining maps. The hyperparameter $\beta$ for the softmax operator is fixed to be $5.0$ for all maps.

### E.1.1 Multi-agent Particle Environments

Table 2 summarizes detailed information for our tested environments based on the multi-agent particle framework with discrete action space, where observation dim and action dim correspond to dimension of observation space and action space for the agents respectively. The world environment is a variant of simple_world_comm, where we modify the environment to allow for Discrete action space instead of MultiDiscrete action space. In world, unmovable entities also include forests (which are accessible) and foods besides the inaccessible landmarks. Figure 5 shows the illustration of the multi-agent particle tasks.

Table 2: Information of environments in the multi-agent particle framework.

| Name | #Agents | #Adver-saries | #Land-marks | State dim | Observation dim | Action dim |
|---|---|---|---|---|---|---|
| Predator-prey (PP) | 3 | 1 | 2 | 16 | 62 | 5 |
| Physical deception (PD) | 2 | 1 | 2 | 10 | 28 | 5 |
| World (W) | 4 | 2 | 1 | 34 | 200 | 5 |
| Covert communication (CC) | 2 | 1 | 2 | 8 | 20 | 4 |

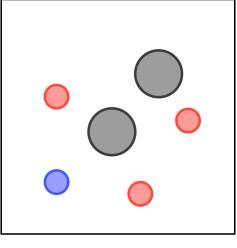 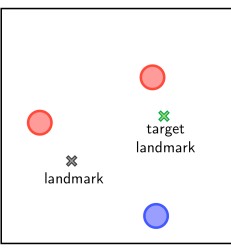 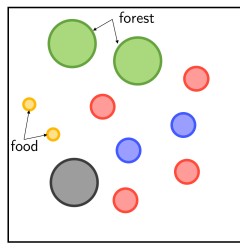 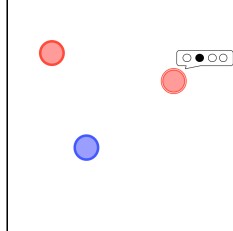

(a) Predator-prey.  (b) Physical deception.  (c) World.  (d) Covert communication.

Figure 5: Illustration of the multi-agent particle tasks.

### E.1.2 StarCraft Multi-Agent Challenge (SMAC)

Table 3 describes the tested maps in SMAC, and Figure 6 shows the illustration for the maps.

Table 3: Information about agents, enemies and difficulty of tested maps in SMAC.

| Name | Agents | Enemies | Difficulty |
|---|---|---|---|
| 3m | 3 Marines | 3 Marines | Easy |
| 2s3z | 2 Stalkers and 3 Zealots | 2 Stalkers and 3 Zealots | Easy |
| 3s5z | 3 Stalkers and 5 Zealots | 3 Stalkers and 5 Zealots | Easy |
| 2c_vs_64zg | 2 Colossi | 64 Zerglings | Hard |
| MMM2 | 1 Medivac, 2 Marauders, and 7 Marines | 1 Medivac, 3 Marauders, and 8 Marines | Super hard |

### E.2 Mean Normalized Return in Multi-Agent Particle Environments

Figure 7 shows the mean normalized return of different algorithms averaged over all multi-agent particle environments. As shown, RES-QMIX significantly outperforms state-of-the-art methods in performance and efficiency.

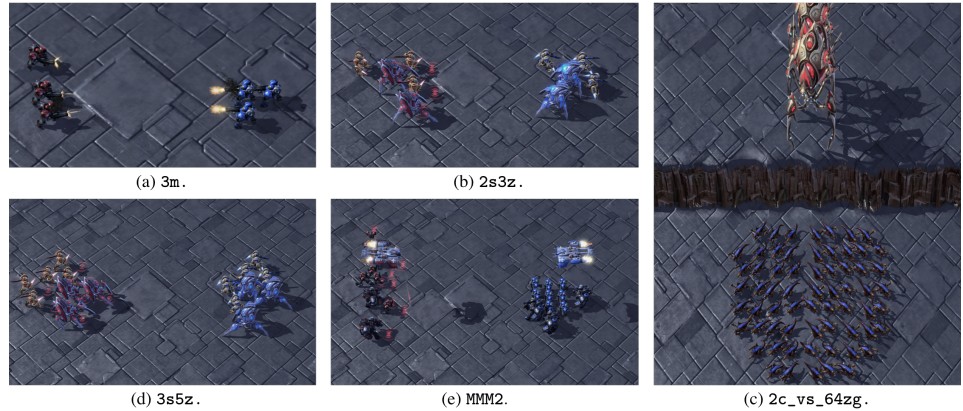

(a) 3m.      (b) 2s3z.

(d) 3s5z.      (e) MMM2.      (c) 2c_vs_64zg.

Figure 6: Illustration of the StarCraft II micromanagement tasks.

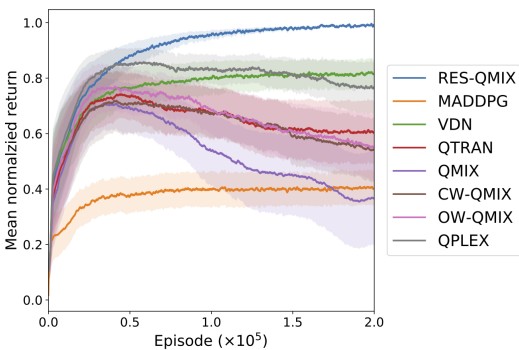

Figure 7: Mean normalized return averaged over all multi-agent particle environments.

## E.3 Performance of RES-QMIX in Stochastic Environments

In this section, we investigate the robustness of RES-QMIX in the stochastic environment. To support stochasticity in the environment, we conduct evaluation in a variant of the predator-prey task based on sticky actions [10], which has been widely used in Atari games [1] to inject stochasticity to the environments. Specifically, at each timestep, the action selected by the agent will be executed with probability $1 - p$, and with probability $p$ the last action taken by the agent will be executed (where the stickiness $p$ is typically set to be $0.25$ in [10]).

Figure 8 shows the performance of RES-QMIX in predator-prey with sticky actions in different levels. As shown, RES-QMIX is robust to different levels of stochasticity and significantly outperforms QMIX in all cases.

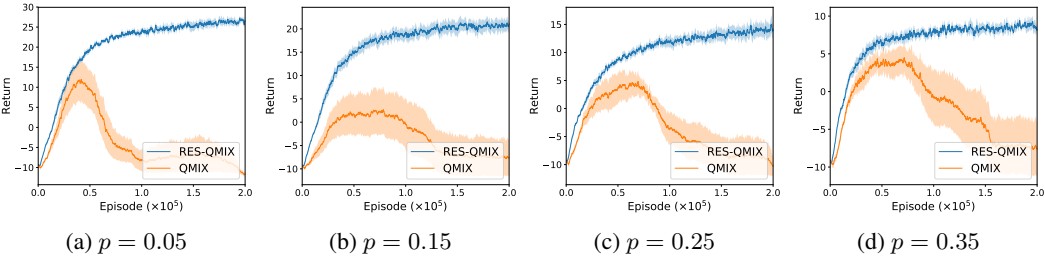

(a) $p = 0.05$    (b) $p = 0.15$    (c) $p = 0.25$    (d) $p = 0.35$

Figure 8: Performance comparison of RES-QMIX and QMIX in stochastic variants of the predator-prey task.

## E.4 Comparison of Value Estimates in Predator-Prey

Figure 9 shows comparison of true values and estimated values (which are obtained in the same way as in Section 4 in the main text) of different algorithms in predator-prey. As discussed in the main text, value estimates for QMIX and Weighted QMIX increase rapidly, which leads to large overestimation bias (Figure 6 in the main text) and severe performance degradation (Figure 5(a) in the main text). Value estimates of VDN (which is based on a linear decomposition of the joint-action $Q$-function) increase more slowly at the end of training, but still incurs large overestimation bias as in QTRAN and QPLEX. Unlike all other value factorization methods, MADDPG learns an unfactored critic that directly conditions on the full state and joint action. It is less sample efficient, which indicates that value factorization is important in these tasks. Thus, MADDPG results in a lower return (Figure 5(a) in the main text) and value estimates compared to all other value factorization methods, but still overestimates. RES-QMIX achieves the smallest bias and fully mitigates the overestimation bias of QMIX, resulting in stable performance and outperforming all other methods (as shown in Figure 5(a) in the main text).

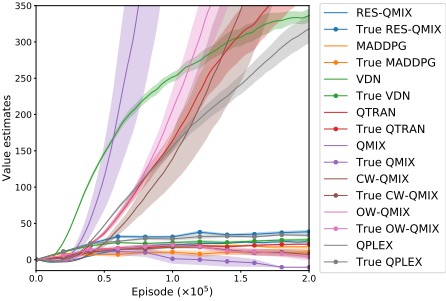

Figure 9: Comparison of true values and estimated values of different algorithms in predator-prey.

## E.5 Full Ablation Study

Figures 10 shows the ablation study of the effect of each component in our approach in different environments by comparing RES-QMIX, RE-QMIX, S-QMIX, and QMIX. As shown, the regularization component is critical for stability, while combining with our softmax operator further improves learning efficiency (or avoids performance oscillation).

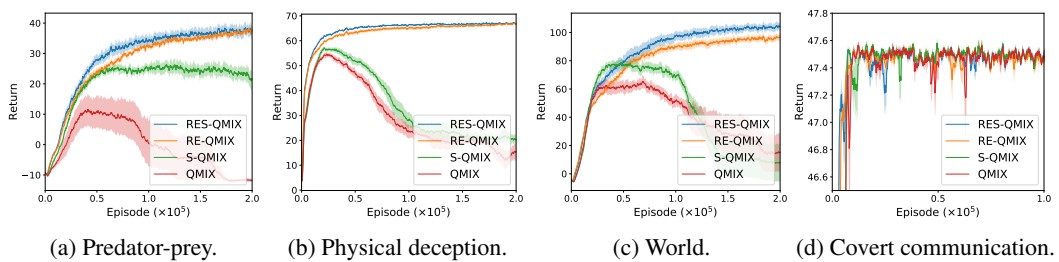

(a) Predator-prey.        (b) Physical deception.        (c) World.        (d) Covert communication.

Figure 10: Ablation study of the effect of each component.

## E.6 Full Learning Curves of RES-QMIX/Weighted QMIX/QPLEX

Our RES method is general and can be readily applied to different $Q$-learning based MARL algorithms. To demonstrate its versatility, we apply it to three different deep multi-agent $Q$-learning algorithms: QMIX, Weighted QMIX, and QPLEX. Full comparison of learning curves of RES over QMIX, Weighted QMIX, and QPLEX in the multi-agent particle environments are shown in Figure 11, 12, and 13, respectively. As shown, our RES-based methods significantly outperform corresponding baseline methods.

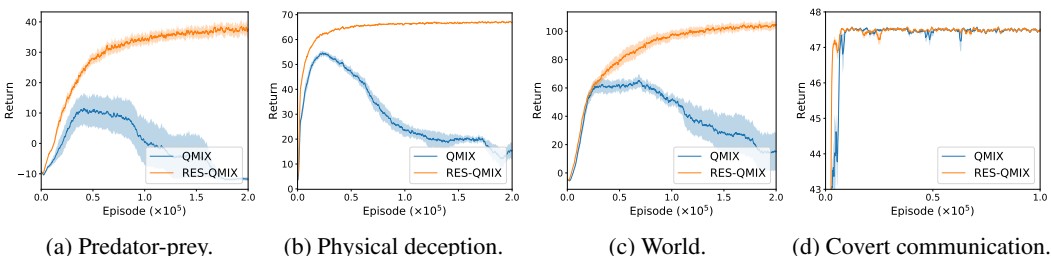

(a) Predator-prey.  (b) Physical deception.  (c) World.  (d) Covert communication.

Figure 11: Performance comparison of QMIX and RES-QMIX.

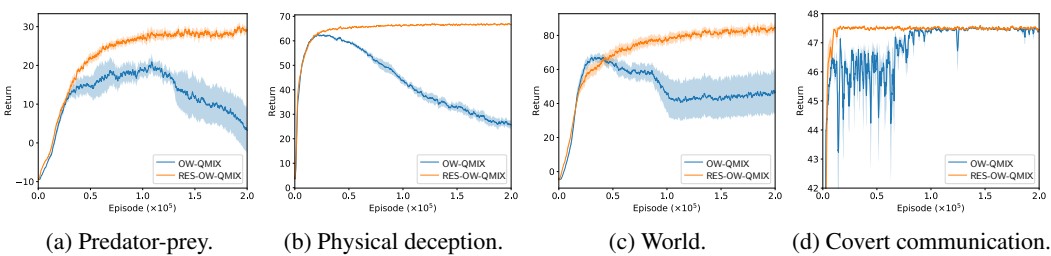

(a) Predator-prey.  (b) Physical deception.  (c) World.  (d) Covert communication.

Figure 12: Performance comparison of Weighted QMIX and RES-Weighted QMIX.

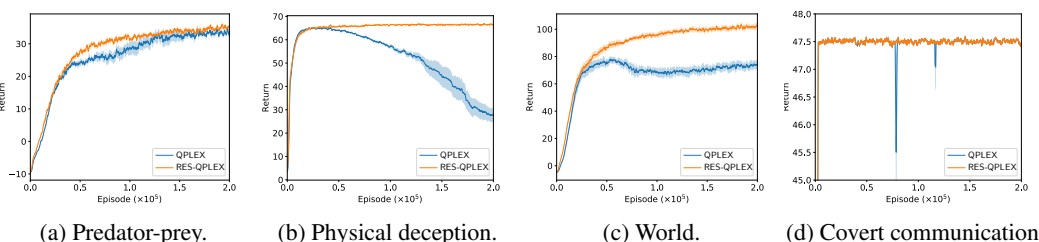

(a) Predator-prey.  (b) Physical deception.  (c) World.  (d) Covert communication.

Figure 13: Performance comparison of QPLEX and RES-QPLEX.

### E.7 Performance Comparison with SM2

Gan et al. [3] propose the soft Mellowmax (SM2) operator to tackle overestimation in reinforcement learning. Figures 14 (a)-(d) show the comparison results of RES-QMIX, SM2-QMIX (with fine-tuned hyperparameters) and QMIX in each environment, and Figure 14(e) summarizes the mean normalized return. As shown, SM2-QMIX fails to tackle the severe overestimation problem, and also significantly underperforms RES-QMIX.

### E.8 Performance Comparison of RES-QMIX with Other Baseline Method in SMAC

In the main text, we analyze how much of a performance improvement RES-QMIX achieves over QMIX in StarCraft II micromanagement tasks in Figure 9. We also compare RES-QMIX in StarCraft II micromanagement tasks against QPLEX, which is the most competitive algorithm in multi-agent particle environments. Experimental settings are the same as in Section E.1.

Figure 15 shows the test win rate, where RES-QMIX significantly outperforms QPLEX in final test win rate and sample efficiency in all but one environments. The only exception is the easy map 3m, where QPLEX is more sample efficient but underperforms RES-QMIX at the end of training. Specifically, the final test win rate is $98.4\%$ and $97.1\%$ for RES-QMIX and QPLEX in 3m respectively. It is also worth noting that QPLEX uses more parameters due to the multi-head attention module for the hypernetwork compared with RES-QMIX and QMIX.

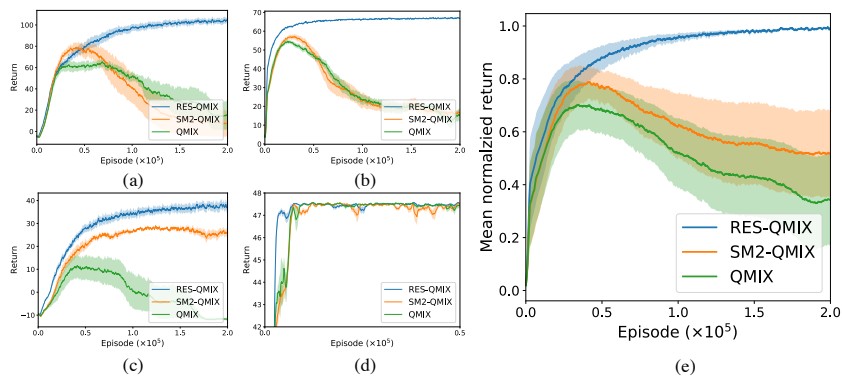

Figure 14: Performance comparison of RES-QMIX, SM2-QMIX and QMIX. (a) Predator-prey. (b) Physical deception. (c) World. (d) Covert communication. (e) Mean normalized return.

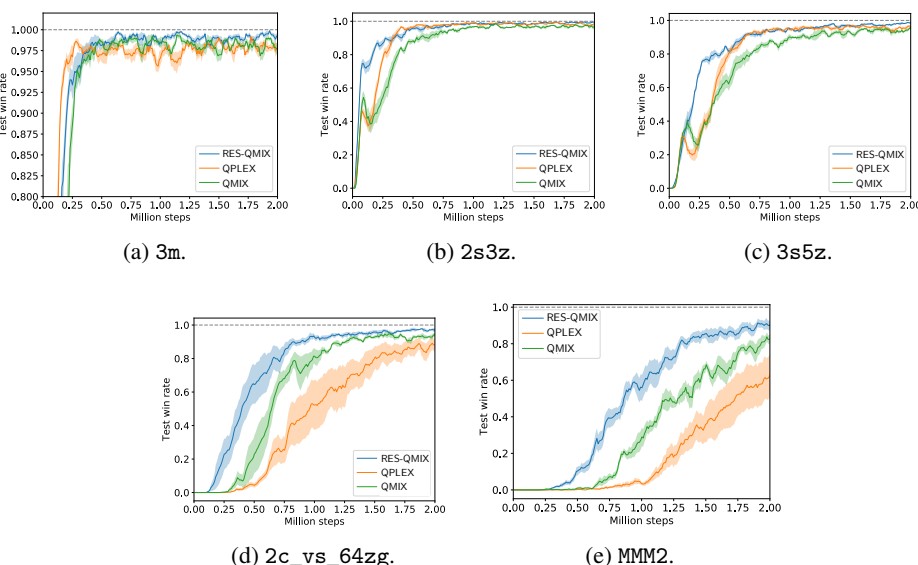

Figure 15: Comparison of test win rate of QMIX, RES-QMIX and QPLEX.

### E.9 Performance of RES-QMIX with Single Estimator in SMAC

Comparison of test win rate of RES-QMIX (single), QMIX (single), and QMIX is shown in Figure 16. Experimental setup is the same as in Section E.1, with the only exception of removing double estimators, i.e., it does not estimate the target value using double estimators as in Double DQN [21]. Environment-specific hyperparameters for RES-QMIX (single) in StarCraft II micromanagement tasks include $\lambda$ and $\beta$. For the super hard map MMM2, $\lambda$ is $1e-1$ while being $5e-2$ for the remaining maps. The parameter $\beta$ is $0.5$ for 3m, $5$ for 2s3z, 3s5z and MMM2, and $50$ for 2c_vs_64zg. For RES-QMIX (single), the computation of the softmax operator is $\mathrm{sm}_{\beta,\hat{U}}(\bar{Q}_{tot}(s',\cdot))$ following Eq. (1) in the main text.

From Figure 16, we can see that RES-QMIX (single) significantly outperforms QMIX and QMIX (single), demonstrating that our RES method is still effective even without double estimators.

## F  Discussion of the SM2 Operator in [3]

A recent work [3] proposes the soft Mellowmax (SM2) operator to tackle overestimation in reinforcement learning, and show that it can be applied to cooperative MARL under the CTDE paradigm.

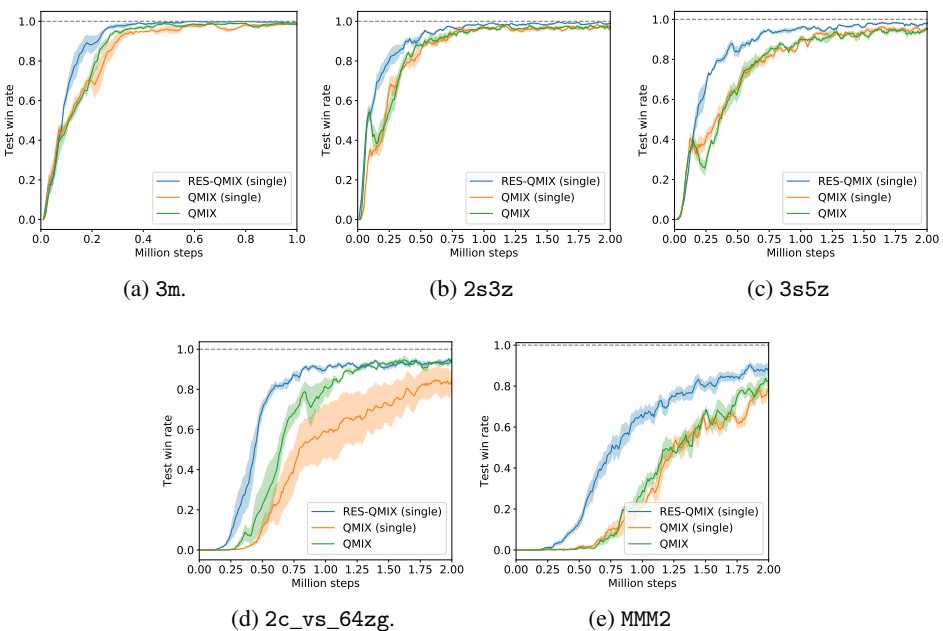

Figure 16: Performance comparison of RES-QMIX (single), QMIX (single), and QMIX.

However, its theoretical analysis relies on a key assumption (Assumption 4.1 in [3]) that the gradients of the monotonic mixing network $f_s$ over agent-wise utilities $Q_a$ are bounded, i.e., $0 \leq l \leq \frac{\partial f_s}{\partial Q_a} \leq L$, $\forall a \in [1, n]$. As shown in Figure 3 in the main text, the gradients $\frac{\partial f_s}{\partial Q_a}$ can increase rapidly and continuously during training, this assumption thus may not hold in practice. In addition, the analysis in Theorems 4.1 and 4.2 in [3] also relies on the assumption that "all the true optimal agent-wise utilities are equal at $Q_a^*(s, u) = V_a^*(s)$". However, the optimal agent-wise utilities $Q_a^*$ are not well-defined in [3]. Please note that there can be an infinite number of possible combinations for $Q_a^*$ and $f_s^*$ to obtain $Q_{tot}^*$. In Figure 14 in Section E.7, we can also see that the SM2 operator is not sufficient and ineffective to solve the severe overestimation problem, and RES-QMIX significantly outperforms SM2-QMIX.

## G  Societal Impacts

In multi-agent reinforcement learning (MARL), an agent interacts with an environment and other learning agents. This is a powerful paradigm that can model many real-world systems, such as autonomous driving [2] and multi-robot exploration [12]. Despite these successes, it is particular important to guarantee the robustness of learning behavior or each agent in the multi-agent system for deploying it in practical problems. Otherwise, the system can collapse. In this paper, we find that the critical overestimation problem in reinforcement learning can be an even more severe practical challenge in MARL than previously acknowledged, and solutions in the single-agent setting fails to successfully tackle this problem. We propose RES to tackle the important and challenging overestimation problem in deep multi-agent $Q$-learning. Our method has the potential to be applied to real-world multi-agent applications, which provides robust value estimates and significantly reduces overestimation bias which can lead to catastrophic performance degradation.