# OpenReview forum: "Regularized Softmax Deep Multi-Agent Q-Learning"
_NeurIPS.cc/2021/Conference — NeurIPS 2021 Poster_

### Official Review · Reviewer_FuxW · 2021-07-14

**Rating:** 7
**Confidence:** 4

**Summary:**

Overestimation in Q-learning is an important problem. However, there has been less attention on the overestimation problem in multi-agent setting compared to the single-agent setting. The authors show that Q-learning in multi-agent setting suffers from overestimation even more than the previously acknowledged. This paper proposes a regularized update scheme that tackles this issue. Also, the new softmax operator, Regularized Softmax (RES) is introduced. Empirical results on various domains, including Starcraft II tasks, are given to support the claim.

**Limitations And Societal Impact:**

I don't see any potential negative societal impact of this work.

**Main Review:**

- The introduction part was very well written. Overestimation is a big challenge in Q-learning, and has been investigated very much by RL community; however, overestimation in multi-agent setting has been less investigated. The authors show that the traditional approaches tackling overestimation in single-agent setting, such as double estimator, resolve the overestimation in multi-agent settings, hence the need for another approach for tackling overestimation in multi-agent settings.
- Background (Dec-POMDP, Deep multi-agent Q-learning, CTDE) was very helpful for readers.
- The argument made in the introduction is extended in the section 3, where the authors did experiments on tackling overestimation using clipped double q-learning and gradient regularization (and the combination of both), and still show that these approaches are not sufficient. Good flow of logic.
- Next approach was to introduce a baseline into the penalty --> \lambda (Q_tot(s,u) - b(s,u))^2. \lambda is a regularization coefficient. N-step return or discounted return can be a baseline.
- The next approach to further resolve overestimation bias caused by max operator was to introduce another softmax operator for multi-agent setting. This softmax operator contains inverse temperature parametere \beta just like a boltzmann softmax. Depending on the choice of \beta, the operator either converges to the max or the mean.
- I think these two step approaches (baseline and softmax) were quite original and novel.
- I've checked most of the proofs, and couldn't find any flaws.
- Multiple Experiments were given in the section 5.
- Authors provided the analysis on the effect of parameters, including regularization coefficient \lambda and \beta. It is quite surprising that the hyperparameter sensitivity of \beta is quite low.

Questions

Q1. The authors mentioned that RES_QMIX is insensitive to the inverse temperature parameter \beta, and set \beta=0.05. Could you give me a more intuition on how this happens? (Usually, the hyperparameter sensitivity and the choice of the optimal parameter is a big issue in softmax operators). Are there any experiments other than the one given in the Figure 7 single domain?

Q2. Seems that the performance of RES-QMIX is quite sensitive to the choice of regularization coefficient labmda. Can you think of any adaptive or automated method to tune the parameter without exhaustive grid-search? Have you ever considered a method to schedule the parameter lambda along the training? or learn to learn the parameter as in the meta-gradient RL (Xu, van Hasselt, Silver et al)?


Suggestions for references:

I'd like the authors add the following citation on the multi-agent RL and overestimation in the related work:

[1] Fu,  Z.,  Zhao,  Q.,  and Zhang,  W.  Reducing overestimation in value mixing for cooperative deep multi-agent reinforcement learning. ICAART, 2020.

Also, the following works that handle overestimation bias in Q-learning (more recent works):

[2] R.Zhu, M. Rigotti. Self-correcting Q-learning. AAAI 2021.

[3] S. Kim, K. Asadi, M. Littman, G. Konidaris. DeepMellow: Removing the Need for a Target Network in Deep Q-learning. IJCAI 2019.

[4] Z. Zhang, Z. Pan, M.J. Kochenderfer. Weighted Double Q-learning. IJCAI 2017

**Time Spent Reviewing:**

9

---

> ### Author Response · Authors · 2021-08-10
> **To Reviewer FuxW**
>
> Thank you for your detailed evaluation of our paper and thoughtful reviews, and the comments are greatly appreciated!
>
> *Q1: The paper shows that RES-QMIX is quite insensitive to $\beta$. Could authors give more intuition on how this happens? Are there any experiments on other tasks besides Figure 7?*
>
> - Low sensitivity of RES-QMIX w.r.t. $\beta$:
>
>   - In the literature, the softmax operator is usually sensitive to the value of $\beta$​. For S-QMIX (a variant of our method that uses the softmax operator in QMIX described in the ablation study section and Figure 7c), its performance is sensitive to different values of $\beta$​. For simplicity of presentation, we summarize in Table 1 the mean normalized return (defined as $R_{norm}=\frac{R - R_{\min}}{R_{\max} - R_{\min}}$​ for each environment) of S-QMIX averaged over all multi-agent particle environments including predator-prey, physical deception, world, and covert communication (we will include the full figures in the revision). As shown, S-QMIX is sensitive to the choice of $\beta$​, since it solely relies on the softmax operator for overestimation reduction.
>
>     |        $\beta$​         | S-QMIX ($0.0005$​) | S-QMIX ($0.005$​) | S-QMIX ($0.05$​) | S-QMIX ($0.5$​)  |
>     | :--------------------: | :---------------: | :--------------: | :-------------: | :-------------: |
>     | Mean normalized return |  $0.539\pm0.364$​  | $0.469\pm0.337$​  | $0.524\pm0.410$​​ | $0.371\pm0.374$​​​​​ |
>
>     Table 1. Comparison of the mean normalized return of S-QMIX at the end of training averaged over all 4 multi-agent particle environments with different values of $\beta$.
>
>   - As for RES-QMIX, we find that the regularization component is critical for stability, while combining with our softmax operator further improves learning efficiency and performance. Different from previous works discussed in the related work section, because RES-QMIX does not solely rely on the softmax operator to reduce overestimation (which is insufficient in the multi-agent setting as discussed in Section 3), the hyperparameter sensitivity of $\beta$​​ is quite low and setting it to be a fixed value works reasonably fine for different tasks.
>
>     |        $\beta$         | RES-QMIX ($0.0005$) | RES-QMIX ($0.005$) | RES-QMIX ($0.05$) | RES-QMIX ($0.5$) |
>     | :--------------------: | :-----------------: | :----------------: | :---------------: | :--------------: |
>     | Mean normalized return |   $0.971\pm0.037$   |  $0.974\pm0.031$   |  $0.987\pm0.015$  | $0.970\pm0.025$  |
>
>     Table 2. Comparison of the mean normalized return of RES-QMIX at the end of training averaged over all 4 multi-agent particle environments with different values of $\beta$.
>
> - Experiments for different values of $\beta$​ on other tasks: Table 2 summarizes the mean normalized return of RES-QMIX averaged over all multi-agent particle environments with different values of $\beta$​​​. The conclusion holds for other tasks, and we will include the full figures for all tasks in the revision.
>
> *Q2: Can authors think of any adaptive method to tune/schedule/learn the regularization coefficient parameter, e.g., meta-gradient RL (Xu et al., 2018)?*
>
> Thank you for your suggestion. Applying the meta-gradient RL method in (Xu et al., 2018) is a promising direction to study adaptive scheduling for the regularization coefficient. It will be an interesting direction for future work.
>
> *Q3: Suggestions for references.*
>
> Thanks for your suggestion for the references on MARL and overestimation. We will definitely cite and discuss these papers in the revision.

---

### Official Review · Reviewer_QGZD · 2021-07-16

**Rating:** 4
**Confidence:** 4

**Summary:**

The author proposes a solution to overestimation in MARL, which is called RES deep Q-learning. The author penalizes large joint-action Q values by two approaches: a regularizer towards discounted return, and an estimated softmax operator. The author also constructs an action subspace for efficient softmax approaximation.

**Limitations And Societal Impact:**

The limitation can be referred to “Main Review” and there is nearly no potential negative societal impact of this work.

**Main Review:**

Generally, I don’t feel very excited after a thorough understanding of this paper. The two main works of the model, mentioned in Summary, are similar to previous methods. For example, the GradReg is a regularization-based method, and softmax operator has been widely study in single-agent RL settings. Though sufficient experimental results are presented, the overall paper is not convincing due to the following problems:
(1)	There’s no significant novelty in this paper. The author just transfers existing solutions in traditional RL to MARL, adding some tricks for efficient computation.
(2)	QMIX suffers from severe overestimation, because the model performance decreases rapidly in four environments after millions of episodes. However, it seems that the overestimation disappears in StarCraft II micromanagement tasks, as the win rate increases without significant degradation (in Figure. 9). Is overestimation a problem with all MARL models in all environments? What is the significance of studying the overestimation problem of specific MARL algorithm in specific environments?
(3)	More theoretical analysis is needed, and experimental results are not quite intuitive. Why those solutions to overestimation in single RL settings fail in MARL? It’s difficult to have an intuitive inspiration without any theoretical analysis. It is also confusing that an experimental result (Figure 1) is given in Introduction without any description in details.
(4)	How is the RES method applied to other algorithms, e.g. weighted QMIX and QPLEX? More details for description is needed.


**Time Spent Reviewing:**

15

---

> ### Author Response · Authors · 2021-08-10
> **To Reviewer QGZD**
>
> Thank you for your comments. Please find our clarification to your questions and concerns below.
>
>
> *Q1: Authors just transfer existing solutions in traditional RL to MARL, e.g., GradReg.*
>
> We emphasize that our main contribution is three-fold. Firstly, we demonstrate that overestimation presents a more severe practical challenge in MARL than previously acknowledged. Solutions in the single-agent domain fail to successfully address this challenge. Secondly, to tackle this important problem, we propose a general method with a novel regularization-based update scheme as well as an efficient and reliable approximation to the softmax operator, which significantly differs from existing literature in single-agent RL. Thirdly, we provide theoretical guarantees for our method and conduct extensive experiments to demonstrate its effectiveness and versatility. The results show that our method significantly outperforms state-of-the-art methods.
>
> Specifically, our method is novel and significantly differs from existing solutions in the following ways:
>
> (1) GradReg aims to avoid overly large value estimates by limiting the gradients from the mixing network of the popular QMIX algorithm as detailed in Section 3. Please note that GradReg is not a core part of the novel contribution of our paper, and our RES method does not use GradReg. We propose GradReg and present it *only* to show that this indirect method fails to tackle the challenging overestimation problem in MARL, which further emphasizes the difficulty and challenge of value estimation in MARL.
>
> (2) We conduct a detailed analysis demonstrating how and why methods from single-agent RL fail to work in Section 3. In particular, popular solutions such as Double DQN and Clipped Double $Q$-learning (from TD3) take an indirect approach, which we show is insufficient in our setting. Instead, we propose a novel way to *directly* penalize large joint-action $Q$-values with a baseline and provide theoretical justification for the overestimation reduction (Theorems 2 and 3).
>
> (3) To further mitigate overestimation bias in the joint-action $Q$​-function due to the max operator, we propose a novel approximation of the softmax operator that is efficient and reliable. We provide theoretical guarantees (Theorem 1) for our new softmax operator, and show that it also achieves better performance (in addition to improving computational efficiency) with minimal additional computation cost. This is important because it is challenging to directly compute the softmax given the exponentially-sized joint action space. Please find the detailed description of the challenge and key differences of our new softmax operator in lines 195-228 and Appendix B.2, and its empirical results in Figure 7c.
>
>
> *Q2.1: Is overestimation a problem with all MARL models in all environments? What is the significance of studying the overestimation problem of a specific MARL algorithm in specific environments?*
>
> Tackling overestimation is a very important problem for MARL algorithms. We propose a general RES method to tackle the challenging overestimation problem for a general class of deep multi-agent $Q$-learning algorithms instead of a specific MARL algorithm. We demonstrate its effectiveness in a variety of standard and widely-used MARL benchmarks instead of specific environments.
>
> - The importance of the overestimation problem:
>   - Tackling overestimation is a general and particularly important problem in RL, and has been extensively studied in the single-agent case. The accumulated errors due to overestimation can lead to sub-optimal policy updates and behaviors, and a number of related works aim to reduce the overestimation bias in order to learn more accurate value estimates. As discussed in the paper (mainly Sections 1 and 3), the problem can be more challenging in MARL. However, the multi-agent setting has received comparatively little attention.
>   - As discussed in lines 23-34, the max operator in $Q$-learning approximates the maximum expected value with the maximum estimated value, which can lead to overestimation for all multi-agent $Q$-learning algorithms. In addition, from the proof in Theorem 3 (Appendix D), we also find that a popular deep multi-agent $Q$-learning algorithm, QMIX, does overestimate.
>
> - The significance of studying the overestimation problem in MARL (in terms of algorithm and environment):
>   - Firstly, we emphasize that we study a general class of MARL algorithms, deep multi-agent $Q$-learning (discussed in Section 2), which includes popular state-of-the-art MARL algorithms such as VDN, QMIX, QTRAN, Weighted QMIX, QPLEX, etc. We show in the paper that these methods *all* suffer from overestimation (Figure 6), and our RES method is generally applicable to $Q$​​​-learning based algorithms, e.g., QMIX, Weighted QMIX, and QPLEX.
>   - Secondly, we evaluate our method in a variety of standard and popular MARL benchmarks and show consistent and significant performance improvement, instead of only studying a specific environment.
>   - Thirdly, our extensive results demonstrate that tackling overestimation is also a critical dimension for improving the performance of MARL algorithms.
>
>
> *Q2.2: Because the model performance decreases rapidly in four environments, QMIX suffers from severe overestimation. The overestimation problem disappears in SMAC, as the win rate does not degrade.*
>
> Please note that there is no direct equivalence between performance degradation and overestimation. For example, it is shown that DDPG can overestimate the value function in (Fujimoto et al., 2018), but the performance of DDPG does not degrade in a number of standard continuous control MuJoCo tasks. However, it significantly underperforms TD3, where TD3 achieves state-of-the-art performance by mitigating the overestimation problem of DDPG. It is important to note that although it is not always the case that performance degrades, the overestimation error can result in sub-optimal policy updates and learning dynamics (Thrun et al., 1993, Fujimoto et al., 2018). In addition, the overestimation problem exists due to the max operator as discussed above and in the paper. Our RES method, which reduces the overestimation bias, provides consistent and significant improvement in SMAC, demonstrating that tackling overestimation is a critical dimension for improving learning efficiency and performance.
>
>
> *Q3.1: Why do those solutions to solve overestimation in single-agent RL settings fail in MARL?*
>
> This is carefully discussed in Section 3, where our analysis shows that utilizing clipped double estimators (a key technique in TD3) as in QMIX (CDQ) is not sufficient for preventing the overly large value estimates in the joint-action Q-function $Q_{tot}$. A key takeaway from our detailed analysis is that these methods take an indirect approach to reducing overestimation in $Q_{tot}$, which we demonstrate is not sufficient. We instead propose a novel way to directly penalize large joint-action $Q$​-values to tackle the problem.
>
>
> *Q3.2: It is confusing to have an experimental result in the introduction without detailed description.*
>
> We present Figure 1 in the introduction as a motivating example, which shows that QMIX can suffer from a more challenging overestimation in practice than previously acknowledged. Its detailed description is in Section 3, with the same experimental setup as in the experimental results section. We will elaborate on more details in the introduction to avoid confusion in the revision.
>
>
> *Q4: How is the RES method applied to other algorithms, e.g., Weighted QMIX and QPLEX?*
>
> Our paper aims to tackle the general class of deep multi-agent Q-learning algorithms in MARL, which includes popular methods including VDN, QMIX, QTRAN, Weighted QMIX, QPLEX, etc. The loss function for this general class of method is introduced in the 2nd paragraph in Section 2. The way to apply the RES method to Weighted QMIX and QPLEX is provided in Eq. (4).

---

### Official Review · Reviewer_9Gqb · 2021-07-17

**Rating:** 7
**Confidence:** 3

**Summary:**

The paper addresses overestimation in return estimator in the context of multi-agent reinforcement learning, while claiming the overestimation could be more severe in multi-agent than in single-agent setting. The authors propose an update scheme to penalize large joint-action Q values that deviate from a baseline through the use of regularizer term in the Bellman update; and include an approximated softmax operator to further improve the overestimation issue. The approximated softmax operator is introduced to reduce computational complexity of employing softmax in multi-agent scenario.

**Limitations And Societal Impact:**

The potential societal impact is discussed in Appendix. However, the authors seem to misinterpret the motive of this section and provided a positive (instead of negative) societal impact.

**Main Review:**

Originality: The authors made clear the difference of its method of applying their proposed approximate softmax operator on Q_tot instead of applying softmax on agent-wise utilities, with the later result available in Appendix. This originality is supported with clear reasoning and theoretical claim of convergence for this approximated softmax operator. The authors also demonstrated the difference of their adopted baseline in the regularizer term from typical implementation of advantage function and made clear to deliberately avoid the presence of Q_tot; instead, the direct discounted return is used.
Quality: The paper establishes the claim of overestimation in QMIX through empirical support in a predator-prey task, in which the value estimators observed to grow without bounds, exploding gradient and weights, causing it to suffer catastrophic performance drop. The authors also provided theoretical formulation to proof reduced trajectory bias using their proposed RES loss function. The authors also examined the proposed method’s sensitivity to hyperparameters and showed better stability. The authors also showed slightly increased runtime of their method compared to QMIX.
Clarity: Paper is well-organized with detailed information of applying the two core components in their proposal: 1) regularizer term and 2) approximated softmax, with the effect of each individual approach is carefully examined. Theoretical justification of reduced bias is also provided for added confidence.
Significance: The authors demonstrated the presence of overestimation bias in MARL (which may be overlooked by many) and achieve significant performance improvement over other MARL methods. They also showed the versatility of their algorithm to be applied to other MARL algorithms other than QMIX. An added value is that the paper is written with a clear thought process of tackling the issue and step-by-step reasoning to apply the two core components could serve a good reference for similar research approaching such issue.


**Time Spent Reviewing:**

4

---

> ### Author Response · Authors · 2021-08-10
> **To Reviewer 9Gqb**
>
> Thank you for your careful reading and detailed evaluation of our paper. We greatly appreciate your positive review!
>
> We will clarify the potential societal impact and have a clearer discussion in the revision.

---

### Decision · Program_Chairs · 2021-09-27

**Decision:**

Accept (Poster)

**Comment:**

The paper addresses an important well-known problem (Q-value overestimation), but in a more context which is more complex than the traditional setup (Multi-Agent RL). In this context the authors claim that the problem is less studied, which the reviewers generally don't contest (and I personally agree as well). This paper studies this problem in this context a clear way, and also propose an algorithm to mitigate it.

Two of the reviewers are voting to accept the paper, whereas the third is recommending a reject. I still recommend acceptance, because that's the majority vote but also because I am not convinced that the third reviewer (QGZD) has a full understanding of the paper. The authors have addressed the criticisms of QGZD and I tend to agree with them. Unfortunately QGZD hasn't answered the rebuttal.